# Subspace Clustering via Tangent Cones

**Amin Jalali**
Wisconsin Institute for Discovery
University of Wisconsin
Madison, WI 53715
amin.jalali@wisc.edu

**Rebecca Willett**
Department of Electrical and Computer Engineering
University of Wisconsin
Madison, WI 53706
willett@discovery.wisc.edu

## Abstract

Given samples lying on any of a number of subspaces, *subspace clustering* is the task of grouping the samples based on the their corresponding subspaces. Many subspace clustering methods operate by assigning a measure of affinity to each pair of points and feeding these affinities into a graph clustering algorithm. This paper proposes a new paradigm for subspace clustering that computes affinities based on the corresponding conic geometry. The proposed *conic subspace clustering (CSC)* approach considers the convex hull of a collection of normalized data points and the corresponding tangent cones. The union of subspaces underlying the data imposes a strong association between the tangent cone at a sample $x$ and the original subspace containing $x$. In addition to describing this novel geometric perspective, this paper provides a practical algorithm for subspace clustering that leverages this perspective, where a tangent cone membership test is used to estimate the affinities. This algorithm is accompanied with deterministic and stochastic guarantees on the properties of the learned affinity matrix, on the true and false positive *rates* and *spread*, which directly translate into the overall clustering accuracy.

## 1 Introduction

Finding a low-dimensional representation of high-dimensional data is central to many tasks in science and engineering. *Union-of-subspaces* have been a popular data representation tool for the past decade. These models, while still parsimonious, offer more flexibility and better approximations to non-linear data manifolds than single-subspace models. To fully leverage union-of-subspaces models, we must be able to determine which data point lies in which subspace. This subproblem is referred to as *subspace clustering* [16].

Formally, given a set of points $x_1, \ldots, x_N \in \mathbb{R}^n$ lying on $k$ linear subspaces $S_1, \ldots, S_k \subset \mathbb{R}^n$, subspace clustering is the pursuit of partitioning those points into $k$ clusters so that all points in each cluster lie within the same subspace among $S_1, \ldots, S_k$. Once the points have been clustered into subspaces, standard dimensionality reduction methods such as principal component analysis can be used to identify the underlying subspaces. A generic approach in the literature is to construct a graph with each vertex corresponding to one of the given samples and each edge indicating whether (or the degree to which) a pair of points could have come from the same subspace. We refer to the (weighted) adjacency matrix of this graph as the *affinity matrix*. An ideal affinity matrix $A$ would have $A(i,j) = 1$ if and only if $x_i$ and $x_j$ are in the same subspace, and otherwise $A(i,j) = 0$. Given an estimated affinity matrix, a variety of graph clustering methods, such as spectral clustering [17], can be used to cluster the samples, so forming the affinity matrix is a critical step.

Many existing methods for subspace clustering with provable guarantees leverage the *self-expressive property* of the data. Such approaches pursue a representation of each data point in terms of the other data points, and then the representation coefficients are used to construct an affinity matrix.

For example, the celebrated *sparse subspace clustering (SSC)* approach of [3] seeks a representation of each sample as a weighted combination of the other points, with minimal $\ell_1$ norm. However, such sparse self-expression can lead to *graph connectivity* issues, e.g., see [10, 8, 20, 5, 19, 13], where clusters can be arbitrarily broken into separate components. This paper proposes a new paradigm for devising subspace clustering algorithms:

> *Conic Subspace Clustering (CSC): exploiting the association of the tangent cones to the convex hull of normalized samples with the original subspaces for computing affinities and subsequent clustering.*

CSC leverages new insights into the geometry of subspace clustering. One of the key effects of this approach is that the learned affinity matrix is generally denser among samples from the same subspace, which in turn can mitigate graph connectivity issues.

In Proposition 1 below, we hint on what we mean by the strong association of the tangent cones with the underlying subspaces for an ideal dataset. In Section 2, we show how a similar idea can be implemented with finite number of samples. Given a set of nonzero samples from a union of linear subspaces, we normalize them to fall on the unit sphere and henceforth assume $X = \{x_1, \ldots, x_N\} \subset \mathcal{S}^{n-1}$ is the set of samples. We further overload the notation to define $X = [x_1, x_2, \ldots, x_N] \in \mathbb{R}^{n \times N}$. *Data hull* refers to the convex hull of samples. The *tangent cone at $x \in \mathrm{conv}(X)$ with respect to* $\mathrm{conv}(X)$ is defined as

$$T(x) := \mathrm{cl}\,\mathrm{conv}\,\mathrm{cone}(X + \{-x\}) = \mathrm{cl}\big\{\textstyle\sum_{x' \in X} \lambda_{x'}(x' - x): \ \lambda_{x'} \geq 0, \ x' \in X\big\}$$

where the Minkowski sum of two sets $A$ and $B$ is denoted by $A + B$, while $A + \{x\}$ may be simplified to $A + x$. The *linear space of a cone* $C$ is defined as $\mathrm{lin}\,C := C \cap (-C)$. We term the intersection of a subspace $S$ with the unit sphere as a *ring* $\mathcal{R} = S \cap \mathcal{S}^{n-1}$.

**Proposition 1.** *For a union of rings, namely $X = (S_1 \cup \ldots \cup S_k) \cap \mathcal{S}^{n-1}$, and for every $x \in X$,*

$$S(x) = \mathrm{span}\{x\} + \mathrm{lin}\,T(x),$$

*where $S(x)$ is the convex hull of the union of all subspaces $S_i$, $i = 1, \ldots, k$, to which $x$ belongs.*

## 1.1 Our contributions

We introduce a new paradigm for subspace clustering, conic subspace clustering (CSC), inspired by ideas from convex geometry. More specifically, we propose to consider the convex hull of normalized samples, and exploit the structure of the tangent cone to this convex body at each sample to estimate the relationships for pairs of samples (to construct an affinity matrix for clustering).

We provide an algorithm which implements CSC (Section 2) along with deterministic guarantees on how to choose the single parameter in this algorithm, $\beta$, guaranteeing no false positives (Section 5) and any desired true positive rate (Section 4), in the range allowed by the provided samples. We specialize our results to random models, to showcase our guarantees in terms of the few parameters defining said random generative models and to compare with existing methods. Aside from statistical guarantees, we also provide different optimization programs for implementing our algorithm that can be used for faster computation and increased robustness (Section 7).

In Section 6, we elaborate on the true positive *rate* and *spread* for CSC and compare it to what is known about a sparsity-based subspace clustering approach, namely sparse subspace clustering, SSC [3]. This comparison provides us with insight on situations where methods such as SSC would face the so called graph connectivity issue, demonstrating the advantage of CSC in such situations.

## 2 Conic Subspace Clustering (CSC) via Rays: Intuition and Algorithm

In this section, we discuss an intuitive algorithm for subspace clustering under the proposed conic subspace clustering paradigm. We present the underlying idea without worrying about the computational aspects, and relegate such discussions to Section 7. All proofs are presented in the Appendix. Henceforth, lower case letters represent vectors, while specific letters such as $x$ and $x'$ are reserved to represent columns of $X$, and $x$ is commonly used as the reference point.

Start by considering Figure 1(a) and the point $x \in \mathcal{R} := (S_1 \cup \cdots \cup S_k) \cap \mathcal{S}^{n-1}$ from which all the rays are emanating. Moreover, define $\mathcal{R}_t := S_t \cap \mathcal{S}^{n-1}$ for $t = 1, \ldots, k$, which gives

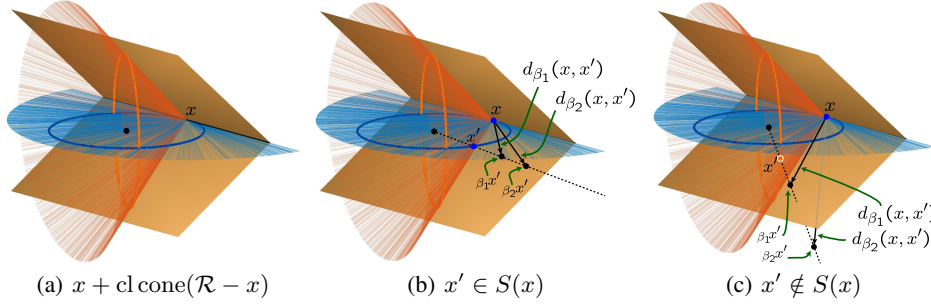

(a) $x + \operatorname{cl}\operatorname{cone}(\mathcal{R} - x)$      (b) $x' \in S(x)$      (c) $x' \notin S(x)$

Figure 1: Illustration of the idea behind our implementation of Conic Subspace Clustering (CSC) *via rays*. The union of the red and blue rings is $\mathcal{R}$, and $x$ is the point from which all the rays are emanating. The orange wedge represents $x + \mathcal{W}(x)$. 1(a) The union of the red and blue *surfaces* is $x + \operatorname{cl}\operatorname{cone}(\mathcal{R} - x)$. 1(b) When $x'$ and $x$ are from the same subspace, the points $d_\beta(x, x')$ for different values of $\beta \geq 0$ lie within $\operatorname{cl}\operatorname{cone}(\mathcal{R} - x)$ – specifically, in the blue shaded cone associated with the blue ring. 1(c) When $x'$ and $x$ are from different subspaces, the points $d_\beta(x, x')$ lie outside $\operatorname{cl}\operatorname{cone}(\mathcal{R} - x)$ for large enough values of $\beta$.

$\mathcal{R} = \mathcal{R}_1 \cup \ldots \cup \mathcal{R}_k$. Only two subspaces are shown and the reference point $x$ is in $\mathcal{R}_1$. The thin red and blue rays correspond to elements of $x + \operatorname{cone}(\mathcal{R} - x) = x + \operatorname{cone}(\{x' - x : x' \in \mathcal{R}\})$, where $\operatorname{cone}(A) := \{\lambda y : y \in A, \lambda \geq 0\}$.[1] We leverage the geometry of this cone to determine subspace membership. Specifically, Figure 1(b) considers a point $x' \in \mathcal{R}_1$ different from $x$. The dashed line segment represents points $-\operatorname{sign}(\langle x, x' \rangle)\beta x'$ for different values of $\beta \geq 0$; where $\operatorname{sign}(0)$ can be arbitrarily chosen as $\pm 1$. The vectors emanating from $x$ and reaching these points represent

$$d_\beta(x, x') := -\operatorname{sign}(\langle x, x' \rangle)\beta x' - x. \tag{1}$$

For $x, x' \in \mathcal{R}_1$, this illustration shows that $d_\beta(x, x') \in \operatorname{cl}\operatorname{cone}(\mathcal{R} - x)$ for any $\beta \geq 0$. In contrast, Figure 1(c) considers $x' \in \mathcal{R}_2$, while $x \in \mathcal{R}_1$. In this case, there exist $\beta > 0$ such that $d_\beta(x, x') \notin \operatorname{cl}\operatorname{cone}(\mathcal{R} - x)$, indicating that $x' \notin S(x)$. Formally,

**Proposition 2.** *For any $x, x' \in \mathcal{R}$ and any scalar value $\beta \geq 0$,*

$$x' \in S(x) \iff \{d_\beta(x, x') : \beta \geq 0\} \subset \operatorname{cl}\operatorname{cone}(\mathcal{R} - x) \tag{2}$$

*Equivalently, $x' \in S(x)$ if and only if $\{\beta \in \mathbb{R} : \beta x' - x \in \operatorname{cl}\operatorname{cone}(\mathcal{R} - x)\}$ is unbounded.*

In other words, we can test whether or not $x' \in S(x)$ by testing the cone membership for $d_\beta(x, x')$. Of course, such a test would not be practical: we cannot compute $d_\beta(x, x')$ for an infinite set of $\beta$ values, the set $\operatorname{cl}\operatorname{cone}(\mathcal{R} - x)$ is generally non-convex (in Figure 1(a), the cone is the union of the red and blue surfaces), and $\operatorname{cl}\operatorname{cone}(\mathcal{R} - x)$ is not known exactly because we only observe a finite collection of points from $\mathcal{R}$ instead of all of $\mathcal{R}$. We now develop an alternative test to (2) that addresses these challenges and can be computed within a convex optimization framework. We first address the convexity issue:

**Proposition 3.** *For the closed convex cone $\mathcal{W}(x) := \operatorname{conv}\operatorname{cl}\operatorname{cone}(\mathcal{R} - x)$, and for any $x, x' \in \mathcal{R}$,*

$$x' \in S(x) \implies \{d_\beta(x, x') : \beta \geq 0\} \subset \mathcal{W}(x). \tag{3}$$

*In other words, $x' \in S(x)$ implies that $\{\beta \in \mathbb{R} : d_\beta(x, x') \in \operatorname{cl}\operatorname{cone}(\mathcal{R} - x)\}$ is unbounded.*

Next, we formulate the test as a convex optimization program, when a *finite* number of samples are given. Specifically, using the samples in $X \subset \mathcal{R}$ instead of all the points in $\mathcal{R}$, we can define an approximation of $\mathcal{W}(x)$ as

$$\mathcal{W}_N(x) := \{(X - x\mathbf{1}_N^T)\lambda : \lambda \in \mathbb{R}_+^N\} \tag{4}$$

which is the *tangent cone (also known as the descent cone)* at $x$ with respect to the data hull $\operatorname{conv}(X)$. The implementation of CSC via rays, as sketched above and detailed below, is based

on testing the membership of $d_\beta(x, x')$ in the tangent cone $\mathcal{W}_N(x)$ for all pairs of samples $x, x'$ to determine their affinity. More specifically, the cone membership test can be stated as a feasibility program, tagged as the *Cone Representability (CR)* program:

$$\min_{\lambda \in \mathbb{R}^N} \ 0 \quad \text{subject to} \quad d_\beta(x, x') = (X - x\mathbf{1}_N^T)\lambda \,, \ \lambda \geq \mathbf{0}_N. \qquad \text{(CR)}$$

If there exists a $\beta \geq 0$ for which (CR) is infeasible, then we conclude $x' \notin S(x)$. Later, in our theoretical results in Sections 4 and 5 we characterize a range (dependent on a target error rate) of possible values of $\beta$ such that for any *single* $\beta$ from this range, checking the feasibility of (CR) for *all* $x, x'$ reveals the true relationships within a target error rate. In Section 7, we discuss a number of variations for the above optimization program. While our upcoming guarantees are all concerned with the cone membership test itself and not the specific implementation, these variations provide better algorithmic options and are more robust to noise. Specifically, we choose to use a variation (in the box below) that is a *bounded feasible linear program* for our implementation of the cone membership test.

We refer to solving any of the variations of the cone membership test for an ordered pair of samples $(x, x')$ and a fixed value of $\beta$ as $\text{CSC}_1(\beta, x, x')$:

> Compute $\widehat{\gamma}(x, x') = \min \{\gamma : (1 - \gamma)(\beta x' - x) = (X - x\mathbf{1}_N^T)\lambda, \ \gamma \geq 0, \ \lambda \geq 0\}$.
> Set $A(x, x') \in \{0, 1\}$ by rounding $1 - \widehat{\gamma}(x, x')$ to either 0 or 1, whichever is closest.

We refer to the optimization program used in the above as the *Robust Cone Membership (RCM)* program. Similarly, solving a collection of these tests for all samples $x'$ and for a fixed $x$, or for all pairs $x, x'$, are referred to as $\text{CSC}_1(\beta, x)$ and $\text{CSC}_1(\beta)$, respectively. When $\text{CSC}_1(\beta)$ is followed by spectral clustering for the constructed affinity matrix, we refer to the whole process as $\text{CSC}(\beta)$. It is worth mentioning that the linear program used in $\text{CSC}_1(\beta, x, x')$ is equivalent to (CR) in a sense made clear in Section 7, and provides the same affinity matrix with a variety of algorithmic advantages, as discussed in Section 7.

## 3 Theoretical Guarantees

In this section, we discuss our approach to providing theoretical guarantees for the aforementioned implementation of CSC via rays. Let us first set some conventions. We refer to a declaration $x' \in S(x)$ (or $x' \notin S(x)$) as *positive* (or *negative*), regardless of the ground truth. Hence, a *true positive* is an affinity of 1 when the samples are from the same subspace, and a *false positive* is an affinity of 1 when the samples are from different subspaces. We provide guarantees for $\text{CSC}_1(\beta)$ to give *no false positives*. This makes the affinity matrix a permuted block diagonal matrix. In this case, if there are *enough well-spread ones in each row* of the affinity matrix, spectral clustering or any other reasonable clustering algorithm will be able to perfectly recover the underlying grouping; see *graph connectivity* in spectral clustering literature [17]. These two phenomena, no false positives and enough well-spread true positives per sample, are the focus of our theoretical results in Sections 4 and 5. In a nutshell, the guarantees boil down to characterizing a range of $\beta$'s for which CSC has controlled degrees of errors: no false positives and a certain true positive rate per row. We also examine the distribution of true positives recovered by our method and illustrate a favorable spread.

Through the intuition behind the cone membership test, namely (CR), it is easy to observe that the number of true positives and the number of false positives are monotonically non-increasing in $\beta$ (which can be observed in Figure 2 as well). Hence, to have a high number of true positives we need to use an upper bounded $\beta$, and to have a few number of false positives we need to use a lower bounded value of $\beta$.

To assess the strength of our deterministic results, we assume probabilistic models on the subspaces and/or samples and study the ranges of $\beta$ for which $\text{CSC}_1(\beta)$ has controlled errors of both types, with high probability. For the random models, we take the number of subspaces to be fixed, namely $k$. However, $\text{CSC}_1(\beta)$ need not know the number of subspaces and spectral clustering can use the gap in the eigenvalues of the Laplacian matrix to determine the number of clusters; e.g., see [15]. In the *random-sample model*, we assume $k$ subspaces are given and samples from each subspace are drawn uniformly at random from the unit sphere on that subspace. In the *random-subspace* model, each subspace is chosen independently and uniformly at random with respect to the Haar measure.

## 3.1 Examples

In this section, we illustrate the performance of the CSC method on some small examples. First, we examine the role of the parameter $\beta$ in $\text{CSC}_1(\beta, x, x')$ and its effect on the false positive and true positive rates in practice. In the first experiment, we have $k = 5$ subspaces, each with dimension $d = 5$, in an $n = 10$ dimensional space, and we draw 30 samples from each of the $k$ 5-dimensional subspaces. We then run $\text{CSC}_1(\beta, x, x')$ for a variety of values of $\beta$ between one and six over 15 random trials; In Figures 2(b), 2(c), and 2(d), we show the results of each trial in thin lines and the means across trials in thick lines (Figure 2(c) shows the median). Figure 2(a) shows, for each value of $\beta$, the histogram of true positive rates across rows. Superimposed on this histogram plot is the empirical mean of the histogram (green solid line) and our theoretical bound from Theorem 6 (purple dashed curve corresponding to the purple dashed curve in Figure 2(b)): for each value of $\beta$, the true positive rate will be above this curve, with high probability.

Our theoretical bounds correspond to sufficient but not necessary conditions. While we observe the tightness of the theory for minimum per-row true positive rate in relation to $\beta$, the wide distribution of per-row true positive rates above the theoretical bound (Figures 2(a) and 2(b)), as well as the spectral clustering step, provide us with good error rates (Figure 2(c)) outside the range of $\beta$'s for which we have guarantees (Figure 2(d)).

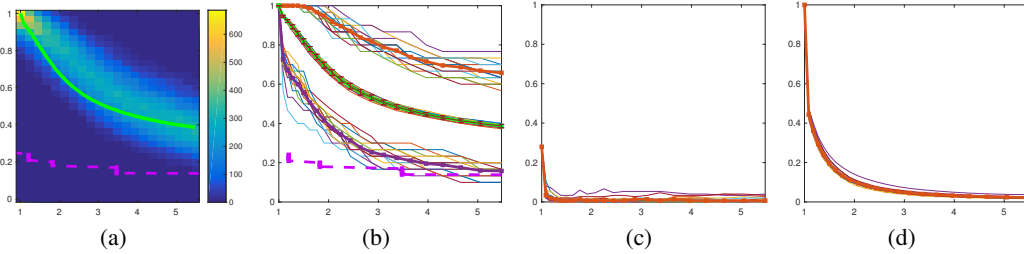

Figure 2: Illustration of the role of $\beta$ (horizontal axis) in determining (a) the histogram of true positive rates across rows, with the empirical mean (solid line) and the theoretical bound (dashed curve, corresponding to the dashed curve in (b)), (b) maximum, mean, and minimum (across rows of the affinity matrix) true positive rates along with the theoretical bound (dashed curve), (c) the clustering mismatch rate after performing spectral clustering, and, (d) the false positive rate. This experiment is in a 10 dimensional space, with 30 random samples from each of 5 random 5-dimensional subspaces, over 15 random trials. Bold curves correspond to averages across trials in (a), (b), and (d), but to the median in (c).

Next, we look at learned affinity matrices output by the proposed CSC method and SSC [3], which is a widely-used benchmark and the foundation of much current subspace clustering research. As described at length in Section D, the true positive rate of SSC is necessarily bounded because of the $\ell_1$ regularization used to learn the affinity matrix. This is not true of CSC – in fact, $\beta$ can be used to control the true positive rate (in an admissible range) as long as it exceeds some lower bound ($\beta \geq \beta_L$). The difference between the true positive rates of SSC and CSC are illustrated in Figure 3. In this experiment, CSC naturally outputs a 0/1 affinity matrix, while the affinity matrix of SSC has a broader diversity of values. We show this matrix and a thresholded version for comparison purposes, where the threshold is set to correspond to a 5% false positive rate.

## 4 Guarantees on True Positive Rates

We study conditions under which a *fraction* $\rho \in (0, 1)$ of samples $x' \in S(x)$ are declared as such. As discussed before, the number of true positives is non-increasing in $\beta$. Therefore, we are interested in an upper bound $\beta_{U,\rho}$ on $\beta$ so that $\text{CSC}_1(\beta, x)$ for $\beta \leq \beta_{U,\rho}$ returns at least $\rho N_t$ true positives ($N_t$ is the number of samples from $S_t$) for any $x \in X^t := X \cap S_t$ and $t = 1, \dots, k$. Consider $\{x\}^\perp := \{y : \langle x, y \rangle = 0\}$. For a close convex set $A$ containing the origin, denote by $\mathbf{r}(A)$ the radius of the largest Euclidean sphere in $\text{span}(A)$ that is centered at the origin and is a subset of $A$.

**Theorem 4** (Deterministic condition for any true positive rate). *The conic subspace clustering algorithm at $x$ with parameter $\beta$, namely $\text{CSC}_1(\beta, x)$, returns a ratio $\rho \in (0, 1)$ of relationships between*

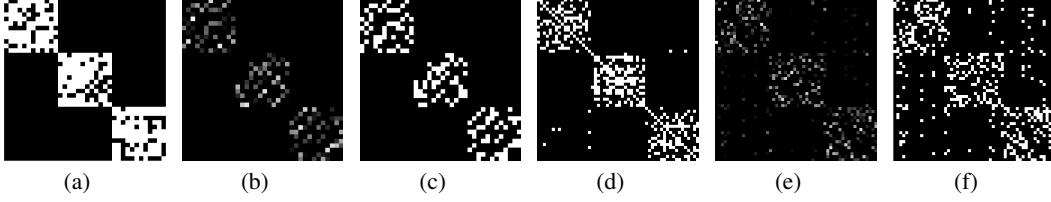

<table>
<tr><td>(a)</td><td>(b)</td><td>(c)</td><td>(d)</td><td>(e)</td><td>(f)</td></tr>
</table>

Figure 3: Affinity matrices for two toy models in an ambient dimension $n = 12$. (a-c) $k = 3$ subspaces, each of rank $d = 4$ and each with $3d = 12$ samples. (a) Result of $\text{CSC}_1(\beta)$. (b) Result of SSC. (c) Thresholded version of (b). (d-e) $k = 3$ subspaces, each of rank $d = 6$ and each with $3d = 18$ samples. (d) Result of $\text{CSC}_1(\beta)$. (e) Result of SSC. (f) Thresholded version of (e), with threshold set so that the false positive rate is $5\%$. As predicted by the theory, CSC achieves higher true positive rates than SSC can.

$x \in X^t$ and other samples as true positives, provided that $\beta \leq \beta_{U,\rho}^x$ where

$$\beta_{U,\rho}^x := \frac{\sin^2(\theta_t^x)}{\cos(\theta_t^x) - \cos(\tilde{\theta}_t^x)} \tag{5}$$

in which, for $m := \lceil \rho(N - 1) \rceil$, $\cos(\tilde{\theta}_t^x)$ is the $(m + 1)$-st largest value among $|\langle x, x' \rangle|$ for $x' \in X^t$, and for $\mathbf{r}(\cdot)$ denoting the inner radius, $\theta_t^x := \arctan(\mathbf{r}((x + \mathcal{W}_N^t(x)) \cap \{x\}^\perp))$. Then, $\text{CSC}_1(\beta)$ is guaranteed to return a fraction $\rho$ of true positives per sample provided that $\beta \leq \beta_{U,\rho} := \min_{x \in X} \beta_{U,\rho}^x$.

As it can be seen from the above characterization, $\theta_t^x$ and $\beta_{U,\rho}^x$ can vary from sample to sample even within the same subspace. When samples are drawn uniformly at random from a given subspace (the random-sample model), the next theorem provides a uniform lower bound on the inner radius and $\theta_t^x$ for all such samples. Note that $\beta_{U,\rho}^x$ is non-decreasing in $\theta_t^x$ and non-increasing in $\tilde{\theta}_t^x$.

**Theorem 5.** *Under a random-sample model, and for a choice $p_t \in (0, 1)$, with probability at least $1 - p_t$, a solution $\theta$ to*

$$\frac{(\cos \theta)^{d_t - 1}}{6\sqrt{d_t} \sin \theta} = \frac{\log(N_t/p_t)}{N_t}$$

*is a lower bound on $\theta_t^x$, which is defined in Theorem 4 and is a function of the inradius of a base of the $t$-th cone $\mathcal{W}_N^t(x)$.*

Theorem 5 is proved in the Appendix using ideas from *inversive geometry* [1]. In a random-sample model, we can quantify the aforementioned $m$-th order statistic. Therefore, we can explicitly compute the upper bound $\beta_{U,\rho}$ (with high probability) in terms of quantities $d_t$ and $N_t$. The final result is given in Theorem 6. Note that both the inradius and the $m$-th order statistic are random variables defined through the samples, hence are dependent. Therefore, a union bound is used.

**Theorem 6.** *Under a random-sample model, $\text{CSC}_1(\beta, x)$ for any $x \in X^t$ yields a fraction $\rho$ of true positives with high probability, provided that $\beta \leq \beta_{u,\rho}^x$, where $\beta_{u,\rho}^x$ is computed similar to (5), using the lower bound on $\theta_t^x$ from Theorem 5 and $\tilde{\theta}_t^x = \frac{\pi}{2}(\frac{m}{N} + \Delta)$. The probability is at least $I(\frac{m}{N} + \Delta; m, N - m) - p_t$, where $I(\cdot; \cdot, \cdot)$ denotes the incomplete Beta function.*

## 5 Guarantees for Zero False Positives

In this section, we provide guarantees for $\text{CSC}_1(\beta, x)$ to return no false positives, in terms of the value of $\beta$. Specifically, we guarantee this by examining a lower bound $\beta_L$ for $\beta$ in $\text{CSC}_1(\beta, x)$. For a fixed column $x$ of the data matrix $X$, we will use $x'$ as a pointer to any other column of $X$. Recall $d_\beta(x, x')$ from (1) and consider

$$\beta_L(x) := \inf \{\beta \geq 0 : d_\beta(x, x') \notin \mathcal{W}_N(x) \ \forall x' \notin S(x)\} \tag{6}$$
$$= \sup \{\beta \geq 0 : d_\beta(x, x') \in \mathcal{W}_N(x) \text{ for some } x' \notin S(x)\}.$$

If the above value is finite, then using any value even slightly larger than this would declare any $x' \notin S(x)$ correctly as such, hence *no false positives*. However, the above infimum may not exist for a general configuration. In other words, there might be a sample $x' \notin S(x)$ for which $d_\beta(x, x') \in \mathcal{W}_N(x)$ for all values of $\beta \geq 0$. The following condition prohibits such a situation.

**Theorem 7** (Deterministic condition for zero false positives). *For $x \in X$, without loss of generality, suppose $S(x) = S_1 \cup \ldots \cup S_j$ for some $j < k$. Provided that all of the columns of $X$ that are not in $S(x)$ are also not in $\mathcal{W}_N(x)$, then $\beta_L(x)$ in (6) is finite. This condition is equivalent to $S_t \cap \mathcal{W}_N(x) = \{0\}$ for all $t = j + 1, \ldots, k$ and all $x \in X \setminus S_t$. In case this condition holds for all $x \in X$, we define*

$$\beta_L := \max_{x \in X} \beta_L(x). \tag{7}$$

*If this condition is met and $\beta \geq \beta_L$ is used, then $\mathrm{CSC}_1(\beta)$ will return no false positives.*

We note that the condition of Theorem 7 becomes *harder* to satisfy as the number of samples *grow* (which makes $\mathcal{W}_N(x)$ larger). While this is certainly not desired, such an artifact is present in other subspace clustering algorithms. See the discussion after Theorem 1 in [11] for examples.

Next, we specialize Theorem 7 to a random-subspace model. Under such model, for $t = j + 1, \ldots, k$, $S_t$ and $\mathcal{W}_N(x)$ are two random objects and are *dependent* (all samples, including those from $S_t$, take part in forming $\mathcal{W}_N(x)$, hence the orientation and the dimension of $S_t$ affect the definition of $\mathcal{W}_N(x)$), which makes the analysis harder. However, these two can be *decoupled* by massaging the condition of Theorem 7 from $S_t \cap \mathcal{W}_N(x) = \{0\}$ into an equivalent condition $S_t \cap \mathcal{W}_N^{-t}(x) = \{0\}$ where $\mathcal{W}_N^{-t}(x) = \mathrm{convcone}\{x' - x : x' \notin S_t\}$; see Lemma 10 in the Appendix. Next, the event of a random subspace and a cone having trivial intersection can be studied using the notion of the *statistical dimension* of the cone and the brilliant Gordon's Lemma (*escape through a mesh*) [6]. The statistical dimension of a closed convex cone $C \subset \mathbb{R}^n$ is defined as $\delta(C) := \mathbb{E} \sup_{y \in C \cap \mathcal{S}^{n-1}} \langle y, g \rangle^2 \leq n$ where $g \sim \mathcal{N}(0, I_n)$. Now, we can state the following lemma based on Gordon's Lemma.

**Lemma 8.** *With the notation in Theorem 7, and under the random-subspace model, $\beta_L$ is finite provided that $\delta(\mathcal{W}_N^{-t}(x)) + \dim(S_t) \leq n$ for $t = 1, \ldots, k$.*

Furthermore, for the above to hold, it is sufficient to have $\sum_{t=1}^{k} \dim(S_t) < n$ (Lemma 11 in the Appendix). Under the above conditions, we are guaranteed that a finite $\beta$ exists such that with high probability, $\mathrm{CSC}_1(\beta)$ results in zero false positives. It is easy to compute $\beta_L$ for certain configurations of subspaces. For example, when the subspaces are independent (the dimension of their Minkowski sum is equal to sum of their dimensions) we have $\beta_L = 1$ (Lemma 13 in the Appendix). Independent subspaces have been assumed before in the subspace clustering literature for providing guarantees; e.g., [2, 3]. Also see [21] and references therein. However, it remains as an open question how one should compute this value for more general configurations. We provide some theoretical tools for such computation in Appendix B.5.

Finally, if $\beta_L$ does not exceed $\beta_{U,\rho}$ from above, then $\mathrm{CSC}_1(\beta)$ successfully returns a (permuted) block diagonal matrix with a density of ones (per row) of at least $\rho$. This allows us to have a good idea about the performance of the post-processing step (e.g., spectral clustering) and hence $\mathrm{CSC}(\beta)$.

## 6 True Positives' Rate and Distribution

Because sparse subspace clustering (SSC) relies upon sparse representations, the number of true positives is inherently limited. In fact, it can be shown that SSC will find a representation of each column $x$ as a weighted sum of *columns that correspond to the extreme rays of* $\mathcal{W}_N(x)$; as shown in Lemma 17 in the Appendix. This phenomenon is closely linked to the graph connectivity issues associated with SSC, mentioned before. In particular, under a random-sample model, the true positive rate for SSC will go to zero as $N_t/d_t$ grows, where $N_t$ is the number of samples from $S_t$ with $\dim(S_t) = d_t$. In contrast, the true positive behavior for CSC has several favorable characteristics. First, if the subspaces are all independent, then the true positive rate $\rho$ can approach one. Second, in unfavorable settings in which the true positive rate is low, it can be shown that the true positives are distributed in such a way that precludes graph connectivity issues (see Section D.3 for more details). Specifically, in the random-sample model, for each subspace $S_t$, there is a matrix $A_{\mathrm{sub}}$

defined below, whose support is contained within the true positive support of the output of $\mathrm{CSC}(\beta)$ for $\beta \in (\beta_L, \beta_{U,\rho})$. Let $X_t$ have i.i.d. standard normal entries, and $\epsilon$ be the $m$-th largest element of $|X_t^T X_t|$. Then, $A_{\mathrm{sub}}$ is defined by $(A_{\mathrm{sub}})_{i,j} = |X_t^T X_t|$ when $|X_t^T X_t| > \epsilon$, and zero otherwise. The distribution of $A_{\mathrm{sub}}$ when columns of $X_t$ are drawn uniformly at random from the unit sphere ensures that graph connectivity issues are avoided with high probability as soon as the true positive rate $\rho$ exceeds $O(\log N_t / N_t)$. As a result, *even if the values of $\rho$ which provide $\beta_{U,\rho} > \beta_L$ are small, there is still the potential of perfect clustering.* These distributional arguments cannot be made for sparsity-based methods like SSC. We refer to Appendix D for more details.

## 7 CSC Optimization and Variations

In Table 1, we provide a number of optimization programs that implement the cone membership test. These formulations possess different computational and robustness properties. Let us introduce a notation of equivalence. We say an optimization program $P$, implementing the cone membership test, is in CR-class if the possible set of its optimal values can be divided into two *disjoint* sets $O_{\mathrm{in}}$ and $O_{\mathrm{out}}$ corresponding to whether $d_\beta(x, x') \in \mathcal{W}_N(x)$ or $d_\beta(x, x') \notin \mathcal{W}_N(x)$, respectively. Then we write $[\![P : O_{\mathrm{in}}, O_{\mathrm{out}}]\!]$; e.g., $[\![(\mathrm{CR}) : 0, \text{infeasible}]\!]$. All of the problems in Table 1 are in CR-class.

Table 1: Different formulations for the cone membership (second column) with their set of outputs when $d_\beta(x, x') \in \mathcal{W}_N(x)$ (third column) and when $d_\beta(x, x') \notin \mathcal{W}_N(x)$. In all of the variations, $A = X - x\mathbf{1}_N^T$, $y$ and $b = d_\beta(x, x')$ live in $\mathbb{R}^n$, and $\lambda$ lives in $\mathbb{R}^N$.

| Tag | Formulation | $O_{\mathrm{in}}$ | $O_{\mathrm{out}}$ |
|-----|-------------|-------------------|--------------------|
| P1 | $\min_\lambda \; 0 \quad \text{s.t.} \quad b = A\lambda \,, \; \lambda \geq 0$ | $\{0\}$ | infeasible |
| P2 | $\min_y \; \langle y, b \rangle \quad \text{s.t.} \quad y^T A \geq 0$ | $\{0\}$ | unbounded |
| P3 | $\min_y \; \langle y, b \rangle \quad \text{s.t.} \quad y^T A \geq 0 \,, \; \langle y, b \rangle \geq -\epsilon$ | $\{0\}$ | $\{-\epsilon\}$ |
| P4 | $\min_{\gamma, \lambda} \; \gamma \quad \text{s.t.} \quad (1-\gamma)b = A\lambda \,, \; \gamma \geq 0, \lambda \geq 0$ | $\{0\}$ | $\{1\}$ |

The first optimization problem (P1) is merely the statement of the cone membership test as a linear feasibility program and (P2) is its Lagrangian dual. (P2) looks for a certificate $y \in \mathcal{W}_N^\star(x)$ (in the dual cone) that rejects the membership of $d_\beta(x, x')$ in $\mathcal{W}_N(x)$. However, neither (P1) nor (P2) are robust or computationally appealing. Next, observe that restricting $y$ to any set with the origin in its relative interior yields a program that is in CR-class. (P3) is defined by augmenting (P2) with a linear constraint, which not only makes the problem bounded and feasible, but the freedom in choosing $\epsilon$ allows for controlling the bit-length of the optimal solution and hence allows for optimizing the computational complexity of solving (P3) via interior point methods. Furthermore, this program can be solved approximately, up to a precision $\epsilon' \in (0, \epsilon)$, and provides the same desired set of results: *an $\epsilon'$-inexact solution for (P3) has a nonnegative objective value if and only if $d_\beta(x, x') \in \mathcal{W}_N(x)$.* If we dualize (P3) and divide the objective by $-\epsilon$ we get (P4) which can also be derived by hand-tweaking (P1). However, the duality relationship with (P3) is helpful in understanding the dual space and devising efficient optimization algorithms. Notice that $(\gamma, \lambda) = (1, 0_N)$ is always feasible, and the optimal solution is in $[0, 1]$. The latter property makes (P4) a suitable candidate for noisy setups without modification. Moreover, $[\![(P4) : 0, 1]\!]$, which makes it a desirable candidate as a proxy for $x' \notin S(x)$. We use this program in our experiments reported in Section 5.1.

## 8 Discussions and Future Directions

This paper describes a new paradigm for understanding subspace clustering in relation to the underlying conic geometry. With this new perspective, we design an algorithm, *CSC via rays*, with guarantees on false and true positive rates and spreads, that sidesteps graph connectivity issues that arise with methods like sparse subspace clustering. This paper should be seen as the first introduction to the idea of, and tools for, *conic subspace clustering,* rather than establishing CSC as the new state-of-the-art, and as a means to ignite future work on several directions in subspace clustering. We focus on our novel geometric perspective and its potential to lead to new algorithms by providing a rigorous theoretical understanding (statistical and computational) and hope that this publication will spur discussions and insights that can inform the suggested future work. A cone membership test is just one approach to exploit this geometry and implement the conic subspace clustering paradigm.

**Remaining Questions.** While more extensive theoretical comparisons with existing methods are necessary, many comparisons are non-trivial because CSC reveals important properties of subspace clustering methods (e.g. spread of true positives) that are not understood for other methods. The limited small-scale experiments were simply intended to illustrate these properties.

Our study of the parameter choice is theoretical in nature and beyond heuristics for implementation. But some questions are still open. Firstly, while we have a clear deterministic characterization for $\beta_U$, tighter characterizations would lead to a larger range for $\beta$. In Figure 2(b), such pursuit would result in a new theoretical curve (instead of the current dashed purple curve) that stays closer to the minimum true positive rate across rows (the lowest thick solid curve). On the other hand, outside of the case of independent subspaces, where $\beta_L = 1$, we only have a deterministic guarantee on the finiteness of $\beta_L$ and computing it for the random-sample model is a topic of current research. Therefore, we do not have a guarantee on the non-triviality of the resulting range $(\beta_L, \beta_U)$. However, as observed in the small numerical examples in Section 5.1, as well as in our more extensive experiments that are not reported here, there often exists a big range of $\beta$ with which we can get perfect clustering.

**Extensions.** While the presented algorithm assumes noiseless data points from the underlying subspaces, our intuition and simulations (synthetic and real data) indicate stability towards stochastic noise. Moreover, the current analysis is suggestive of algorithmic variants that exhibit robust empirical performance in the presence of stochastic noise. This is why, similar to advances in other subspace clustering methods, we hope that the analysis for the noiseless setup provides essential insights to provably generalize the method to noisy settings. Furthermore, there remain several other open avenues for exploration, particularly with respect to theoretical and large-scale empirical comparisons with other methods, and extensions to measurements corrupted by adversarial perturbations, with outliers among the data points, as well as with missing entries in the data points. By design, SSC and other similar methods require a full knowledge of data points. CSC imposes the same requirement and an open question is how to extend the CSC framework when some entries are missing from the data points.

## Footnotes

[1]Note that this is not the same as a conic hull.

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
