[Supplementary Material]

# Subspace Clustering via Tangent Cones
# Supplementary Material

**Amin Jalali** (`amin.jalali@wisc.edu`), **Rebecca Willett** (`willett@discovery.wisc.edu`)

Let us fix some notation. For a given set $A$,

- $\operatorname{conv}(A) := \big\{ \sum_{i=1}^{t} \lambda_i z_i : t \in \mathbb{N}, \ z_i \in A, \ \lambda_i \geq 0, \ \sum_{i=1}^{t} \lambda_i = 1 \big\}$ denotes the convex hull of $A$, where $\mathbb{N}$ is the set of positive integers.
- $\operatorname{cone}(A) := \big\{ \lambda y : \ y \in A, \ \lambda \geq 0 \big\}$,
- the conic hull is denoted by $\operatorname{conv cone}(A) = \big\{ \sum_{i=1}^{t} \lambda_i z_i : \ t \in \mathbb{N}, z_i \in A, \lambda_i \geq 0 \big\}$.
- the closure is denoted by $\operatorname{cl} A$.

Minkowski sum of two sets $A$ and $B$ is denoted by $A + B$, while $A + \{x\}$ may be simplified to $A + x$. For a subspace $S$, the orthogonal complement is defined as $S^{\perp} = \{ y : \ \langle y, x \rangle = 0, \ \forall x \in S \}$. We abuse the notation to define $\{x\}^{\perp} = \{ y : \ \langle y, x \rangle = 0 \}$.

Consider $k$ linear subspaces in $\mathbb{R}^n$, namely $S_1, \ldots, S_k$. We term the intersection of a subspace $S_i$ with the unit sphere as a *ring* $\mathcal{R}_i = S_i \cap \mathcal{S}^{n-1}$, for $i = 1, \ldots, k$, and we define $\mathcal{R} = \mathcal{R}_1 \cup \ldots \cup \mathcal{R}_k = (S_1 \cup \cdots \cup S_k) \cap \mathcal{S}^{n-1}$. For any $x \in \mathcal{R}$, denote by $S(x)$ the convex hull of union of all subspaces $S_i$ such that $x \in S_i$. The subspace $S(x)$ can also be described as the direct sum (denoted by $\oplus$) of those subspaces that contain $x$. Moreover, $\mathcal{R}(x) := S(x) \cap \mathcal{S}^{d-1}$.

Given a set of nonzero sample points from a union of linear subspaces, we normalize them to fall on the unit sphere, and without loss of generality, assume $X = \{x_1, \ldots, x_N\} \subset \mathcal{S}^{n-1}$ is the set of samples. We overload the notation to define a corresponding matrix as $X = [x_1, x_2, \cdots, x_N]$. *Data hull* refers to the convex hull of samples,

$$\operatorname{conv}(X) := \big\{ \sum_{i=1}^{t} \lambda_i x_i : \ 0 \leq \lambda_i \leq 1, \ x_i \in X, \ t = 1, \ldots, N \big\}.$$

The *tangent cone at a point* $x \in \operatorname{conv}(X)$ *with respect to* $\operatorname{conv}(X)$ is defined as the closed conic hull of shifted samples,

$$T(x) := \operatorname{cl conv cone}(X - \{x\}) = \operatorname{cl}\big\{ \sum_{i=1}^{t} \lambda_i (x_i - x) : \ \lambda_i \geq 0, \ x_i \in X, \ t = 1, \ldots, N \big\}$$

where the closure operator can be omitted when $x \in X$. The *linear space of a cone* $C$ is defined as

$$\operatorname{lin} C := C \cap (-C).$$

We denote the subset of columns of $X$ which lie on $S_t$ by $X^t \in \mathbb{R}^{n \times N_t}$. Lower case letters represent vectors, while specific letters such as $x$ and $x'$ are reserved to represent columns of $X$, and $x$ is commonly used as the reference point. Lower case greek letters represent scalars or scalar-valued functions.

## A  Details on the CSC Paradigm

Let us list a few preliminary facts regarding $d_\beta(x, x') = -\operatorname{sign}(\langle x, x' \rangle)\beta x' - x$, defined for any $\beta \geq 0$ and any two points $x, x'$ on the unit sphere. We allow $\operatorname{sign}(0)$ to be arbitrarily chosen as $+1$ or $-1$. In Figure 4, $d_\beta(x, x')$ is the vector connecting $x$ to $\beta x'$. Observe that

$$\|d_\beta(x, x')\|_2^2 = 1 + 2\beta|\langle x, x'\rangle| + \beta^2.$$

It is easy to verify that the intersection of $x + \operatorname{cone}(d_\beta(x, x'))$ and the unit sphere, denoted by $x''$, is given by

$$x'' := (1 - a)x - a\beta\operatorname{sign}(\langle x, x'\rangle)x' = x - a\,(x + \beta\operatorname{sign}(\langle x, x'\rangle)x') = x + a\,d_\beta(x, x') \quad (8)$$

where

$$a := \frac{2 + 2\beta|\langle x, x'\rangle|}{1 + 2\beta|\langle x, x'\rangle| + \beta^2} \in (0, 2] \quad (9)$$

and $a$ is a continuously decreasing function of $\beta$ for $\beta > 0$. Moreover, $\|d_\beta(x, x')\|_2^2 = \frac{\beta^2 - 1}{1 - a}$.

Figure 4: An illustration for different quantities considered in Appendix A. Given $x$ and $x'$, the plane of our paper corresponds to the plane $\mathrm{span}\{0, x, x'\}$, and the dotted circle represents the unit circle in this plane. For a fixed value of $\beta > 1$, the intersection of $x + \mathrm{cone}(d_\beta(x, x'))$ and the unit sphere is denoted by $x''$ and is characterized in (8). The triangles $(\beta x', 0, x)$ and $(\beta x', x', x' + \lambda x)$ are similar, in the given order. We define $\lambda$ through similarity of triangles as $\frac{\lambda}{1} = \frac{\beta - 1}{\beta}$ which implies $\beta = \frac{1}{1-\lambda}$.

## A.1 Proof of Proposition 2

*Proof.* First of all, notice that $\mathrm{sign}(\langle x, x' \rangle)\beta x' - x$ always lies outside of $\mathrm{cone}(\mathcal{R} - x)$ for large enough $\beta$ when $\langle x, x' \rangle \neq 0$; because the following quantity,

$$\langle x, \mathrm{sign}(\langle x, x' \rangle)\beta x' - x \rangle = \beta|\langle x, x' \rangle| - 1,$$

can be made positive while $\langle x, x'' - x \rangle = \langle x, x'' \rangle - 1 \leq 0$ for all $x'' \in \mathcal{R}$. Secondly, the case of $x' = -x$ is trivial.

(i) Suppose $x' \in S(x)$. We claim that

$$\left\{ d_\beta(x, x') := -\mathrm{sign}(\langle x, x' \rangle)\beta x' - x : \ \beta \geq 0 \right\} \subset \mathrm{cl}\,\mathrm{cone}(\mathcal{R} - x). \tag{10}$$

where $\mathrm{sign}(0)$ can be arbitrarily chosen as $+1$ or $-1$. For any fixed $\beta \geq 0$, define

$$x'' := (1-a)x - a\beta\mathrm{sign}(\langle x, x' \rangle)x' \in S(x) \ , \ a = \frac{2 + 2\beta|\langle x, x' \rangle|}{1 + 2\beta|\langle x, x' \rangle| + \beta^2} > 0 \tag{11}$$

which, after algebraic manipulations, can be shown to satisfy $x'' \in S(x) \cap \mathcal{R}$ as well as

$$d_\beta(x, x') = \frac{1}{a}(x'' - x) \in \mathrm{cone}(\mathcal{R} - x) \cap S(x)$$

which establishes the claim. Observe that $1 + 2\beta|\langle x, x' \rangle| + \beta^2 = |1 + \beta\exp(i\angle x, x')|^2$.

(ii) Conversely, suppose $\{d_\beta(x, x') : \ \beta \geq 0\} \subset \mathrm{cone}(\mathcal{R} - x) \subset \mathrm{cone}(\mathcal{S}^{n-1} - x)$. Notice that the corresponding point on $\mathcal{S}^{n-1}$ (hence the one on $\mathcal{R}$) is unique: by the assumption, for any $\beta \geq 0$ there exists $\lambda \geq 0$ and $x'' \in \mathcal{S}^{n-1}$ for which $d_\beta(x, x') = \lambda(x'' - x)$. If this is satisfied by two pairs $(x_1, \lambda_1)$ and $(x_2, \lambda_2)$ then $\lambda_1(x_1 - x) = \lambda_2(x_2 - x)$ which implies

$$\|\lambda_1 x_1 - \lambda_2 x_2\|_2^2 = \|(\lambda_1 - \lambda_2)x\|_2^2 \implies \langle x_1, x_2 \rangle = 1 \implies x_1 = x_2.$$

The trajectory of these unique points on $\mathcal{S}^{n-1}$ for all $\beta \geq 0$ is a half-circle from $-x$ (corresponding to $\beta = 0$) to $x'$ (corresponding to $\beta = 1$) to $x$ (corresponding to $\beta$ approaching $+\infty$), and is assumed to be on $\mathcal{R}$. Since $\mathcal{R}$ is a collection of rings, $x$ and $x'$ must on the same subspace.

$\square$

## A.2 Proof of Proposition 3

*Proof.* Similar to what mentioned in the beginning of proof of Proposition 2, $\mathrm{sign}(\langle x, x' \rangle)\beta x' - x$ always lies outside of $\mathrm{cone}(\mathcal{R} - x)$ for large enough $\beta$ when $\langle x, x' \rangle \neq 0$; the argument holds even after taking the convex hull of $\mathrm{cone}(\mathcal{R} - x)$. Moreover, the first part of proof of Proposition 2 applies exactly the same, as $\mathrm{cone}(\mathcal{R} - x) \subset \mathcal{W}(x)$. $\square$

# B  Guarantees for No False Positive Declarations

## B.1  Proof of Theorem 7

*Proof.* Since there are only a finite number $N$ of samples $x'$, it is enough to show that for each $x' \notin S(x)$ there exists a finite value of $\beta \geq 0$ for which $d_{\beta+\varepsilon}(x, x') \notin \mathcal{W}_N(x)$ for any $\varepsilon > 0$. To simplify the presentation, suppose $\text{sign}(\langle x, x' \rangle) = 1$, hence $d_\beta(x, x') = \beta x' - x$.

Suppose the condition of the theorem holds: $x' \notin S(x)$ and $x' \notin \mathcal{W}_N(x)$. By definition, $\beta_L > 0$. First, notice that $\mathcal{W}_N(x)$ is closed, pointed (i.e., contains no linear subspace, as $N$ is finite and there is no infinitesimal sequence of samples on $S(x)$ approaching $x$), and full-dimensional in $\text{span}(X) \subseteq \mathbb{R}^n$. Therefore,

$$
\begin{aligned}
\beta_L(x, x') &:= \inf\{\beta > 0 : \ \beta x' - x \notin \mathcal{W}_N(x)\} \\
&= \sup\{\beta > 0 : \ \beta x' - x \in \mathcal{W}_N(x)\} \\
&= \sup\{\beta > 0 : \ x' - \frac{1}{\beta} x \in \mathcal{W}_N(x)\} \quad (12)
\end{aligned}
$$

is finite. Since the cone is full-dimensional in $\text{span}(X)$, and $x, x' \in \text{span}(X)$, as well as $-x \in \mathcal{W}_N(x)$, the line $\{\alpha : \ x' + \alpha x \in \mathcal{W}_N(x)\}$ is a half line in $\mathbb{R}$; it is not unbounded both ways. We show that this half line only has negative values. Contrapositively, assume that there exists $\alpha \geq 0$ for which $x' + \alpha x \in \mathcal{W}_N(x)$, while $x' \notin \mathcal{W}_N(x) \ni -x$. For a convex set $A$, denote by $\Pi(x, A)$ the orthogonal projection of $x$ onto $A$, namely $\Pi(x, A) := \arg\min_{y \in A} \|x - y\|_2$. Using the definition of polar cones, as well as the Moreau decomposition for cones, $x' + \alpha x \in \mathcal{W}_N(x)$ implies

$$
\begin{aligned}
0 &\geq \langle x' + \alpha x, \Pi(x', \mathcal{W}_N(x)^\circ) \rangle \\
&= \langle \Pi(x', \mathcal{W}_N(x)^\circ) + \alpha x, \Pi(x', \mathcal{W}_N(x)^\circ) \rangle \\
&= \|\Pi(x', \mathcal{W}_N(x)^\circ)\|_2^2 + \alpha \langle x, \Pi(x', \mathcal{W}_N(x)^\circ) \rangle.
\end{aligned}
$$

On the other hand, $-x \in \mathcal{W}_N(x)$ implies $\langle -x, \Pi(x', \mathcal{W}_N(x)^\circ) \rangle \leq 0$. Therefore, from the above equation, we get $\Pi(x', \mathcal{W}_N(x)^\circ) = 0_n$, which implies $x' \in \mathcal{W}_N(x)$ and contradicts the assumption. This, in conjunction with (12), establishes the claim, and $\beta_L(x, x')$ exists. □

Figure 5 is helpful in understanding the condition of Theorem 7. Here, the cone and the subspace only have a trivial intersection. Therefore, Theorem 7 guarantees that there exists a finite value of $\beta$ for which $d_\beta(x, x')$ is outside of the cone, for all $x'$ from the green subspace.

Figure 5: A schematic with samples from two subspaces, and the cone $x + \mathcal{W}_N(x)$ in orange, where $x$ belongs to the set of red samples. The green circle represents the subspace corresponding to the green samples, shifted by $x$.

A list of conditions that imply the condition of Theorem 7 are given in the next lemma. These impose restrictions on the subspaces, rather than the samples, and are easier to check.

**Lemma 9.** *Any of the following conditions, if met for all $i = j + 1, \ldots, k$, implies the condition of Theorem 7:*

- $S_i \cap \mathcal{W}_N(x) = \{0\}$.

- $\Pi(\mathcal{W}_N^\star(x); S_i) = \mathbb{R}^{d_i}$ *(where for a given cone $C$, the dual cone is denoted by $C^\star = -C^\circ$).*

- *For any $\theta$, $(\theta + S_i^\perp) \cap \mathcal{W}_N^\star(x) \neq \emptyset$. Or equivalently, for any $\theta \in S_i$, $(\theta + S_i^\perp) \cap \mathcal{W}_N^\star(x) \neq \emptyset$.*

- $\mathbf{0} \in \mathrm{int}\{U_i^T z : z \in B\} \subset \mathbb{R}^{d_i}$ *where $B$ is any convex base for $\mathcal{W}_N^\star(x)$, and $U_i$ is an orthonormal basis of $S_i$.*

## B.2 Proof of Lemma 8

We specialize Theorem 7 to a random-subspace model by using the celebrated Gordon's lemma [6] and the notion of *statistical dimension*.

Under a random-subspace model, $S_t$ and $\mathcal{W}_N(x)$ are two random objects and are *dependent* (samples from $S_t$ take part in forming $\mathcal{W}_N(x)$, hence the orientation and the dimension of $S_t$ affect the definition of $\mathcal{W}_N(x)$), which makes the analysis harder. Therefore, assuming a symmetric set of samples (or simply considering $[X, -X]$ as the new set of samples), Lemma 10 attempts to *decouple* $S_t$ and $\mathcal{W}_N(x)$. Define $\mathcal{W}_N^i(x) = \mathrm{convcone}\{x' - x : x' \in S_i\}$, with $\mathcal{W}_N(x) = \mathrm{conv} \bigcup_{i=1}^k \mathcal{W}_N^i(x)$. Moreover, consider $\mathcal{W}_N^{-i}(x) = \mathrm{convcone}\{x' - x : x' \notin S_i, x' \in X\}$.

**Lemma 10.** *Suppose $S(x) = S_1 \cup \ldots \cup S_j$ and take any $i = j + 1, \ldots, k$. Then, for a set of symmetric data points denoted by $X$, we have that $S_i \cap \mathcal{W}_N(x) = \{0\}$ and $S_i \cap \mathcal{W}_N^{-i}(x) = \{0\}$ are equivalent.*

*Proof.* As $\mathcal{W}_N^{-i}(x) \subset \mathcal{W}_N(x)$, the condition $S_i \cap \mathcal{W}_N(x) = \{0\}$ simply implies $S_i \cap \mathcal{W}_N^{-i}(x) = \{0\}$. On the other hand, $\mathcal{W}_N(x) = \mathrm{conv}(\mathcal{W}_N^{-i}(x) \cup \mathrm{cone}(X^i - x))$. Suppose there exists a nonzero $y \in S_i \cap \mathcal{W}_N(x)$ while $S_i \cap \mathcal{W}_N^{-i}(x) = \{0\}$. Then, there exists $z \in \mathcal{W}_N^{-i}(x)$, $x' \in X^i$, $\lambda \in [0, 1)$, and $\theta > 0$, for which $y = \lambda z + (1 - \lambda)\theta(x' - x)$.

As $y, x' \in S_i$, we should have $\lambda z - (1 - \lambda)\theta x \in S_i$. On the other hand, we have $z, -x \in \mathcal{W}_N^{-i}(x)$ which implies $\lambda z - (1 - \lambda)\theta x \in \mathcal{W}_N^{-i}(x)$. Therefore, $\lambda z - (1 - \lambda)\theta x \in S_i \cap \mathcal{W}_N^{-i}(x)$ which is a contradiction unless it is the zero vector. But then we should have $\lambda z = (1 - \lambda)\theta x$ implying $\lambda \neq 0$ and $\langle z, x \rangle = (1 - \lambda)\theta/\lambda > 0$ for $z \in \mathcal{W}_N^{-i}(x)$ which contradicts the fact that all samples are on the unit sphere implying $\langle w, x \rangle \leq 0$ for all $w \in \mathcal{W}_N(x)$. $\square$

In a random-subspace model, based on Lemma 10, the condition of Theorem 7 requires the following events to happen simultaneously for $i = j + 1, \ldots, k$: a random subspace $S_i$ of known dimension $d_i < n$ has trivial intersection with a random cone $\mathcal{W}_N^{-i}(x)$ where $S_i$ and $\mathcal{W}_N^{-i}(x)$ are independent. In this scenario, $x$ can be thought of as a fixed point which will not lie on $S_i$ with probability 1 as desired. However, notice that these events for $i = j + 1, \ldots, k$ are not independent from each other (the subspace from one event is dependent on the cone in another event) and we will use a union bound in the end. Since $k$ is not assumed to be growing with $n$ in our problem setup, such a union bound does not require special considerations. For now, we focus on one of the events.

The event of a random subspace and a cone having trivial intersection can be studied using the notion of the statistical dimension of the cone and the brilliant Gordon's Lemma [6]. Let us define some quantities first. Consider two random variables $g \sim \mathcal{N}(0, I_n)$ and $u \sim \mathrm{unif}(\mathcal{S}^{n-1})$. The statistical dimension of a closed convex cone $C \subset \mathbb{R}^n$ is defined as

$$\delta(C) := \mathbb{E} \sup_{y \in C \cap \mathcal{S}^{n-1}} \langle y, g \rangle^2 = n \, \mathbb{E} \sup_{y \in C \cap \mathcal{S}^{n-1}} \langle y, u \rangle^2 = \mathbb{E}\|\Pi(g; C)\|_2^2 = \mathbb{E} \, \mathrm{dist}^2(g, C^\circ) \quad (13)$$

where $C^\circ := \{y : \langle x, y \rangle \leq 0 \; \forall x \in C\}$ is the polar cone. Lemma 8 then simply employs Gordon's Lemma [6], and the proof is straightforward based on the above discussions as well as the fact that the statistical dimension is bounded above by the ambient dimension.

## B.3 A simple sufficient condition for existence of $\beta_L$

**Lemma 11.** *For any configuration of $k$ subspaces in $\mathbb{R}^n$, and any point $x \in \mathcal{S}^{n-1}$ from their union, we have*

$$\delta(\mathcal{W}_N^t(x)) \leq \dim(S_t)$$

*for $t = 1, \ldots, k$.*

*Proof.* Suppose $x \in S_1 \cap \ldots \cap S_j$ and not in the rest of the subspaces $S_{j+1}, \ldots, S_k$. For $t \in \{1, \ldots, j\}$, we have $x \in S_t$ which implies $\mathcal{W}_N^t(x) \subset S_t$. Therefore,

$$\delta(\mathcal{W}_N^t(x)) \leq \delta(S_t) = d_t.$$

For $t \in \{j+1, \ldots, k\}$ and $t \neq i$, we consider the whole ring and use the bound $\mathcal{W}_N^t(x) \subset \mathrm{convcone}(\mathcal{R}_t - x)$. Therefore, by the monotonicity of the statistical dimension, we get

$$\delta(\mathcal{W}_N^t(x)) \leq \delta(\mathrm{convcone}(\mathcal{R}_t - x)). \tag{14}$$

It is known that $\delta(C) + \delta(C^\star) = n$. Hence, we can either find an upper bound for $\delta(\mathcal{W}_N^{-i}(x))$ or a lower bound for $\delta(\mathcal{W}_N^{-i}(x)^\star)$. Therefore,

$$\begin{aligned}
\delta(\mathcal{W}_N^t(x)^\star) &\geq \delta(\mathrm{convcone}(\mathcal{R}_t - x)^\star) \\
&= \mathbb{E} \, \mathrm{dist}^2(g, \mathrm{convcone}(\mathcal{R}_t - x)) \\
&= \mathbb{E} \inf_\theta \|g - U_t\theta + x\|_2^2 \\
&= \mathbb{E}\|g_{S_t^\perp} + x_{S_t^\perp}\|_2^2 + \mathbb{E} \inf_\theta \|U_t^T(g + x) - \theta\|_2^2 \\
&= n - d_t + \|x_{S_t^\perp}\|_2^2 \\
&\geq n - d_t
\end{aligned}$$

Therefore, $\delta(\mathcal{W}_N^t(x)) \leq d_t$. $\qquad\square$

As briefly mentioned right after Lemma 8, while we can compute the statistical dimension for the individual cones, it remains to understand how they should be aggregated, along with the affinity between the subspaces, to provide us with the value of $\delta(\mathcal{W}_N(x))$ in the random model.

We can simplify the condition of Lemma 8 and derive a sufficient condition:

**Lemma 12.** *With the notation in Theorem 7, and under the random-subspace model, $\beta_L$ is finite provided that $\sum_{t=1}^k \dim(S_t) < n$.*

The proof is simply by noting that $\mathcal{W}_N^{-i}(x) \subset \oplus_{j \neq i} S_j \oplus \{x\}$ which implies $\delta(\mathcal{W}_N^{-i}(x)) \leq 1 + \sum_{j \neq i} \dim(S_j)$. Plugging this in to Lemma 8 yields the desired result.

### B.4 $\beta_L$ when subspaces are independent (dimension of sum is the sum of dimensions)

**Lemma 13.** *Given $k$ independent subspaces, $\beta_L = 1$.*

*Proof.* By definition, $\beta_L \geq 1$. We will show that it cannot be larger than one. Consider $x \in X^1$ and $x' \in X^2$, and denote the rest of the points by $\bar{X}$. Suppose $\beta$ is such that $\beta x' - x \in \mathcal{W}_N(x)$. Hence, there exists $\lambda \geq 0$ such that

$$\beta x' - x = (X - x\mathbf{1}^T)\lambda = X^1\lambda_1 + X^2\lambda_2 + \bar{X}\bar{\lambda} - x\mathbf{1}^T\lambda$$

where $X = [X^1, X^2, \bar{X}]$ and $\lambda^T = [\lambda_1^T, \lambda_2^T, \bar{\lambda}^T]$. Assume, without loss of generality, that $(\lambda_1)_x = 0$. By the linear independence assumption, we get

$$x(\mathbf{1}^T\lambda - 1) = X^1\lambda_1 \quad, \quad \beta x' = X^2\lambda_2.$$

Since $\frac{1}{\mathbf{1}^T\lambda_1}X^1\lambda_1$ is a convex combination of points in $X^1$, and $x \notin \mathrm{conv}(X^1\backslash\{x\})$ (as all points are on the sphere), we get $\frac{\mathbf{1}^T\lambda - 1}{\mathbf{1}^T\lambda_1} < 1$ which, due to $\lambda \geq 0$, implies $\mathbf{1}^T\lambda_2 < 1$. Therefore, $\beta x' = X^2\lambda_2 \in \mathrm{conv}(X^2)$ which requires $\beta \leq 1$. This establishes the claim; $\beta_L = 1$. $\qquad\square$

### B.5 More on computing and bounding $\beta_L$ for general subspace configurations

Theorem 7 only guarantees the existence of $\beta_L$. However, computing this extremal value is not straightforward. In the following, we present a transformation of $\beta_L(x, x')$ into a function that is *concave* in $x'$, hence can be maximized over convex sets. Such a property can be used as an

analytical tool in deriving the desired lower bound when certain information (or a probabilistic generative model) about the subspaces is available.

For this, we use a simple geometric argument illustrated in Figure 6. Consider the line segment extended from $x'$ in the direction of $x$ until it reaches the boundary of the cone illustrated with solid black lines. By similarity of triangles $(0, x, \beta x')$ and $(x', x' + \lambda x, \beta x')$, we have $\frac{\lambda}{1} = \frac{\beta - 1}{\beta}$ which yields $\beta = \frac{1}{1-\lambda}$. More generally, we can define

$$\lambda_0(x, x') := \inf \left\{ \lambda \geq 0 : \ x' - (1 - \lambda)x \notin \mathcal{W}_N(x) \right\}$$

as well as $\lambda_0(x) := \max_{x' \notin S(x)} \lambda_0(x, x')$ and $\lambda_0 := \max_{x \in X} \lambda_0(x)$. Then, the above geometric argument can be used to observe that

$$\lambda_0(x, x') = 1 - \frac{1}{\beta_L(x, x')}.$$

With this insight, we turn (6) and (7) into computing the corresponding extremal value of $\lambda_0 = 1 - \frac{1}{\beta_L}$. In our problem, $\lambda_0$ measures how close the rings $\mathcal{R}_{j+1}, \ldots, \mathcal{R}_k$ are to $\mathcal{W}_N(x)$. The above relationship can also be seen from (12), where for $x' \notin \mathcal{W}_N(x)$, we have $1 - \lambda_0(x, x') = \inf\{\alpha : \ x' - \alpha x \in \mathcal{W}_N(x)\}$, and $\alpha$ is a placeholder for $\frac{1}{\beta}$.

Interestingly, $\lambda_0(x, x')$ coincides with the function $\lambda_{\min}(x' - x)$ defined in [12], and we can leverage their optimization approach for maximizing $\lambda_0(x, x')$. More specifically, the definition of $\lambda_{\min}$ in [12] requires a proper closed convex cone with nonempty interior and a reference point in its relative interior. Thus, $\lambda_0(x, x') = \lambda_{\min}(x' - x)$ for $\mathcal{W}_N(x)$ (which is a proper closed convex cone with nonempty interior for every finite $N$) and with reference point $-x$ (if in the relative interior of $\mathcal{W}_N(x)$). By the characterization of [12], $\lambda_0(x, x')$ is a concave function of its second argument which is computationally desirable for computing the maximum over rings to which $x$ does not belong, for all $x$.

Figure 6: Definition of $\lambda$ through similarity of triangles as $\frac{\lambda}{1} = \frac{\beta - 1}{\beta}$ which yields $\beta = \frac{1}{1-\lambda}$. The solid black lines represent $x + \mathcal{W}_N(x)$ and the dotted circle represents the unit sphere, while both of which have been intersected with the plane $\text{span}\{0, x, x'\}$.

## C   Partial True Positives

### C.1   Proof of Theorem 4

We assume each sample only belongs to one of the subspaces. This is not a restrictive assumption, as for example, in a random model, a random sample $x$ will belong to only one subspace, with probability one.

*Goal:* assuming $x \in S_t$, find an upper bound $\beta_{U,\rho}^t$ such that if we use any $\beta \leq \beta_{U,\rho}^t$ in $\text{CSC}_1(\beta, x)$, a fraction $\rho$ of the samples in $S_t$ are true positives. Then, with a

$$\beta \leq \beta_{U,\rho} := \min_{t=1,\ldots,k} \beta_{U,\rho}^t,$$

*every row* of the estimated affinity matrix will have at least $\rho(N_t - 1)$ ones over the block corresponding to samples from $S_t$, excluding the diagonal entries. With this in mind, from this point forward, we may omit all the subscripts $t$ and restrict the space to $S_t$ to simplify the presentation.

Suppose we are given $N-1$ samples (except $x$) in this $d$-dimensional subspace (where we have omitted subscripts form $N_t$ and $d_t$). We will denote the samples, excluding $x$, by $X$. Moreover, suppose we are given a desired *fraction* for true positives, namely $\rho \in (0,1)$. Define the desired *number* of true positives (excluding the diagonal entry) as

$$m := \lceil \rho(N-1) \rceil \in \{1, \ldots, N-1\}.$$

Consider sorting the values $\arccos(|\langle x, x' \rangle|)$ for all $x' \in X$ ($x' \neq x$), namely as

$$0 = \arccos(|\langle x, x \rangle|) < \arccos(|\langle x, x_1 \rangle|) \leq \arccos(|\langle x, x_2 \rangle|) \leq \ldots \leq \arccos(|\langle x, x_{N-1} \rangle|),$$

and taking the $m$-th smallest value and defining

$$\eta_\rho := \arccos(|\langle x, x_m \rangle|) \quad , \quad \epsilon_\rho := |\langle x, x_m \rangle| = \cos(\eta_\rho).$$

Moreover, for any given value $\epsilon \in [0,1]$, denote a spherical cap and its symmetrized version by

$$C_\epsilon^- := \left\{ y \in \mathcal{S}^{d-1} : \langle y, x \rangle \leq -\epsilon \right\} \quad , \quad C_\epsilon := \left\{ y \in \mathcal{S}^{d-1} : |\langle y, x \rangle| \geq \epsilon \right\}.$$

By definition, for every $x' \in X \cap C_{\epsilon_\rho}$ we have $|\langle x, x' \rangle| \geq \epsilon_\rho = |\langle x, x_m \rangle|$. Recall that we are interested in an upper bound for $\beta$ below which we get at least $m$ true positives from $\text{CSC}_1(\beta, x)$ (except $x$ itself). Consider,

$$\beta(x') := \sup \left\{ \beta > 0 : d_\beta(x, x') \in \mathcal{W}_N^t(x) \right\}, \tag{15}$$

where $\mathcal{W}_N^t(x) := \text{convcone}\{x' - x : x' \in S_t\}$, and note that the above quantity might be strictly smaller than a counterpart defined via the $\mathcal{W}_N(x)$ defined with all samples included; but this restricted definition suffices for our purposes. Therefore, our interest is in

$$\beta_1 := \inf \left\{ \beta(x') : x' \in X \cap C_{\epsilon_\rho} \right\}.$$

Using any $\beta \leq \beta_1$, by definition, $\text{CSC}_1(\beta, x)$ would output all $x' \in X \cap C_{\epsilon_\rho}$ as true positives, where from the definition of $\epsilon_\rho$, the set $X \cap C_{\epsilon_\rho}$ contains at least $m \geq \rho(N-1)$ of the samples.

For any sample $x' \in X$, consider the half-plane

$$H(x, x') := \left\{ \nu_1 x + \nu_2 x' : \nu_1, \nu_2 \in \mathbb{R}, \ \nu_2 \cdot \text{sign}(\langle x, x' \rangle) \leq 0 \right\}$$

which has the line $\text{cone}(x)$ as its boundary. These are all the points in the plane defined by $x$ and $x'$ that are on the same side of the line $\text{cone}(x) \cup \text{cone}(-x)$ as $-\text{sign}(\langle x, x' \rangle)x'$. Then, consider a point $y_{x'}$ on the boundary of $C_{\epsilon_\rho}^- \cap H(x, x')$ as

$$y_{x'} := \nu_1 x + \nu_2 x' \in \mathcal{S}^{d-1} \quad \text{where} \quad \nu_2 \cdot \text{sign}(\langle x, x' \rangle) \leq 0 \quad \text{and} \quad \langle y_{x'}, x \rangle = -\epsilon_\rho. \tag{16}$$

It is easy to see that for any $x' \in X \cap C_{\epsilon_\rho}$ we have $\beta(y_{x'}) \leq \beta(x')$, where $\beta(y_{x'})$ is defined as in (19) as a simple adaptation of (15). However, $y_{x'}$ may not be in the cone anymore, which is not relevant for our purposes. Therefore, we can lower bound $\beta_1$ as

$$\beta_2 := \inf \left\{ \beta(y_{x'}) : x' \in X \cap C_{\epsilon_\rho} \right\} \leq \beta_1. \tag{17}$$

For future purposes, we re-state the definition of $\beta_2$ in terms of $y_{x'}$ rather than $x'$. Considering $X \cap C_{\epsilon_\rho} = \{x_1, \ldots, x_m\}$, define $Y_{\epsilon_\rho} = \{y_{x_1}, \ldots, y_{x_m}\}$. Then,

$$\beta_2 = \inf \left\{ \beta(y) : y \in Y_{\epsilon_\rho} \right\} \leq \beta_1. \tag{18}$$

Now, let us define a number of quantities illustrated in Figure 7.

Define

$$\theta(x') := \max\{\angle(-x, y) : y \in \mathcal{W}_N(x) \cap H(x, x')\},$$
$$r(x') := \tan \theta(x').$$

Note that $\theta(x') = \theta(x'')$ for any other $x'' \in (\mathcal{W}_N(x) + x) \cap H(x, x')$. Specifically, $r(y_{x'}) = r(x')$. Moreover, for any $y \in H(x, x') \cap \mathcal{S}^{d-1}$, define

$$\beta(y) := \sup\{\beta > 0 : \beta y \in (x + \mathcal{W}_N(x)) \cap H(x, x')\} = \sup\{\beta > 0 : \beta y \in x + \mathcal{W}_N(x)\}. \tag{19}$$

Figure 7: An illustration of the quantities used in the proof of Theorem 4. The two solid black lines, emanating from $x$, represent $\mathcal{W}_N^t(x)$. For simplicity, we have used an $x'$ with $\langle x, x' \rangle \leq 0$. Here, we use the shorthands $\beta = \beta(y_{x'})$, $r = r(x')$, and $\theta = \theta(x')$.

Define

$$\gamma : \mathbb{R}_+ \times \mathbb{R}_+ \to \mathbb{R} \quad , \quad \gamma(r, \beta) := \frac{\sqrt{r^2\beta^2 - r^2 + \beta^2} - r^2}{\beta(1 + r^2)}$$

which is an increasing function in $\beta$ (fixing $r$) and decreasing function in $r$ (fixing $\beta$); i.e., partial derivatives satisfy $\partial_\beta \gamma(r, \beta) > 0$ and $\partial_r \gamma(r, \beta) < 0$. Now, simple triangle geometry along with the definition in (16) provides us with

$$\gamma(r(x'), \beta(y_{x'})) = \epsilon_\rho \tag{20}$$

and we are interested in a lower bound on $\beta(y_{x'})$; to be able to lower bound $\beta_2$ from (17). Using the monotonicity properties of $\gamma(\cdot, \cdot)$, we will find $\bar{r}(x')$ satisfying $\bar{r}(x') \leq r(x')$ which will provide us with $\bar{\beta}(x')$ satisfying $\bar{\beta}(x') \leq \beta(y_{x'})$ and

$$\gamma(\bar{r}(x'), \bar{\beta}(x')) = \epsilon_\rho$$

where we take the largest solution as $\bar{\beta}(x')$. The above equation is in fact a quadratic equation in $\beta$, with roots equal to (we use $r$ for $\bar{r}(x')$ to declutter)

$$\frac{\epsilon r^2(1 + r^2) \pm r^2\sqrt{1 + r^2}}{1 + r^2 - \epsilon^2(1 + r^2)^2} = \frac{r^2}{\pm\sqrt{1 + r^2} - \epsilon(1 + r^2)}$$

and we denote the larger one by $\bar{\beta}(x')$. An equivalent expression is given by

$$\bar{\beta}(x') = \frac{\sin^2(\theta)}{\cos(\theta) - \epsilon} \quad \text{where} \quad \theta := \arctan(\bar{r}(x')). \tag{21}$$

Note that the expression of $\bar{\beta}(x')$ is increasing in $r = \bar{r}(x')$ (when $\epsilon_\rho$ is small enough): as a result of the monotonicity properties of $\gamma$ and the Equation (20). Therefore, using larger lower bounds for $r(x')$, i.e., tighter ones, provides us with larger values for $\bar{\beta}(x')$ which translates into more freedom in choosing $\beta$ for $\text{CSC}_1(\beta, x)$. All in all,

$$\beta_3 := \inf\{\bar{\beta}(x') : \ x' \in X \cap C_{\epsilon_\rho}\} \leq \beta_2.$$

Note that $\bar{\beta}(x')$ in (21) is an increasing function of $\bar{r}(x')$ and $\theta$. Therefore,

$$\beta_3 = \frac{\sin^2(\theta)}{\cos(\theta) - \epsilon} \quad \text{where} \quad \theta := \arctan(r) \quad \text{and} \quad r = \inf\{\bar{r}(x') : \ x' \in X \cap C_{\epsilon_\rho}\}. \tag{22}$$

Now, we turn into an ingredient given in the statement of Theorem 4, to find $r$ in the above. From the definition of the inner radius and $r(\cdot)$, it is clear that for all $x'$ we have

$$r = \inf\{\bar{r}(x') : \ x' \in X \cap C_{\epsilon_\rho}\}$$
$$\geq \inf\{\bar{r}(x') : \ x' \in \mathcal{S}^{d-1}\}$$
$$= \mathbf{r}\left((x + \mathcal{W}_N^t(x)) \cap \{x\}^\perp\right).$$

This establishes the deterministic part of the proposition. In a nutshell, here is the pipeline

$$\left.\begin{array}{l} \rho \xrightarrow{\text{determines}} m, \epsilon \\ N \xrightarrow{\text{determines}} r, \theta \end{array}\right\} \xrightarrow{\text{determine}} \beta_3 = \beta_{U,\rho}^t. \tag{23}$$

## C.2 Proof of Theorem 5

We first study the inradius, in Sections C.2.1 and C.2.2, without assuming any random generative model. We provide a characterization for the inradius in Lemma 14 as an extremal value related to the dual of the tangent cone. In Section C.2.2, we present a simple approach based on *inversive geometry* to derive a lower bound for $\mathbf{r}((x + \mathcal{W}_N^{1,\dots,j}(x)) \cap \{x\}^\perp)$ that might be easier to compute in different setups we assume later on. These quantities are derived using the *inversion mapping*. In Section E we review some background on the inversion mapping and develop some tools for our purposes. In Section C.2.3, we specialize this result to the random-sample model, to prove Theorem 5.

### C.2.1 A characterization for the inradius

Recall that for a close convex set $A$ containing the origin, the inradius of $A$ denoted by $\mathbf{r}(A)$ is the radius of the largest Euclidean sphere in $\mathrm{span}(A)$ that is *centered at the origin* and is a subset of $A$.

Consider the cone $\mathcal{W} := \mathcal{W}_N^{1,\dots,j}(x)$. The inradius of the base $(x + \mathcal{W}) \cap \{x\}^\perp$ of $\mathcal{W}$ is closely related to the circular cone $C$ with largest opening and central axis $-x$ such that $C \subseteq \mathcal{W}$. For such a cone, we also have $\mathcal{W}^\star \subseteq C^\star$. Therefore, $\mathbf{r}((x + \mathcal{W}) \cap \{x\}^\perp)$ is related to the circular cone with smallest opening that contains $\mathcal{W}^\star$. This intuition in formalized in the next lemma.

**Lemma 14** (deterministic characterization of the inradius)**.** *The inradius*

$$r = \mathbf{r}\left((x + \mathcal{W}_N^{1,\dots,j}(x)) \cap \{x\}^\perp\right)$$

*is given by* $r = \frac{p}{\sqrt{1-p^2}}$ *where*

$$p := \min_y \frac{\langle x, y \rangle}{\|y\|_2} \quad \text{subject to} \quad y^T(X^{1,\dots,j} - x\mathbf{1}^T) \leq 0. \tag{24}$$

*In other words, for the optimal solution* $y^\star$ *we have*

$$r^2 = \frac{\langle x, y^\star \rangle^2}{\|y^\star\|_2^2 - \langle x, y^\star \rangle^2}. \tag{25}$$

The proof is straightforward and is mainly about the relationship between the cosine of largest opening for the dual cone (whose cosine is given by $\frac{\langle x, y^\star \rangle}{\|y^\star\|_2}$), namely $p$, and the inner radius of the original cone, namely $r$.

### C.2.2 Inradius, via inversion

We first provide some intuition on the proof of Theorem 5 without using any randomness assumptions. Consider Figure 8. Consider an inversion map centered at $x$ and with radius $\sqrt{2}$ [1]. By the rules of inversion, the inverse of the black line ($\ell_{AB}$) is a circle (labeled as $I(\ell_{AB})$ and shown as a black circle) that passes through $x$ and the two points on the sphere shown as red dots. Since the circle $\mathcal{C}$, the orange circle centered at $x$ and tangent to the black line, has a zero degree with the black line ($\ell_{AB}$), its inverse, which is another circle centered at $x$ (the smaller orange circle labeled as $I(\mathcal{C})$) will have a zero degree with the black circle ($I(\ell_{AB})$); i.e., they will be tangent. This shows that the radius of the smaller orange circle ($I(\mathcal{C})$) is equal to the diameter of the black circle ($I(\ell_{AB})$). Hence, we need to upper bound the radius of the black circle.

All in all, a lower bound on the desired inradius (see Figure 10) can be derived by upper bounding the radius of circles like the black circle, and considering the *maximum*. The black circle is a circle passing through $x$ and any other two samples *that correspond to an edge of the convex hull* of the radially projected points. A careful reader observes the hardness of the problem, as we cannot simply take the maximum radius.

### C.2.3 Lower bound on the inradius, in a random-sample model, via Lemma 14

In this section, we turn into a random model where samples are drawn independently and uniformly at random from each subspace. Consider the subspace $S_1$ and a sample $x \in S_1$. Since samples are

Figure 8: Part of two equators are shown in blue and they pass through two sample points on the sphere that have been marked red. We are interested in lower bounding the radius of the orange circle in the plane (which represents $\{x\}^\perp$). For this, it is enough to lower bound the radius of the orange circle tangent to the black line and centered at $x$.

drawn at random, we will have $S(x) = S_1$ with probability one. However, we may still use the notation $S(x)$ to make the relationship between $x$ and the subspace to which it belongs more clear throughout.

For a random-sample model, we derive conditions under which the inradius is at least $r_0$ with high probability. Using the terminology of Lemma 14,

$$r \geq r_0 \iff p \geq p_0 := \frac{r_0}{\sqrt{1 + r_0^2}}.$$

Therefore, we are interested in showing that $\mathbb{P}(p \leq p_0)$ is small; possibly as a function of the number of samples $N_1$.

From (24), observe that $p \leq p_0$ is equivalent to

$$\langle y, x \rangle \leq p_0 \quad \text{and} \quad \max_{x' \in X^1} \langle y, x' \rangle \leq \langle y, x \rangle \tag{26}$$

for some $y \in \mathcal{S}^{d-1}$. Then,

$$\mathbb{P}(p \leq p_0) = \mathbb{P}\left( \exists y \in \mathcal{S}^{d-1} \; : \; \langle y, x \rangle \leq p_0 \; , \; \max_{x' \in X^1} \langle y, x' \rangle \leq \langle y, x \rangle \right) \tag{27}$$

$$\leq \mathbb{P}\left( \exists y \in \mathcal{S}^{d-1} \; : \; \langle y, x \rangle = p_0 \; , \; \max_{x' \in X^1} \langle y, x' \rangle \leq \langle y, x \rangle \right) \tag{28}$$

where the last term is the probability that a spherical cap of opening degree $\arccos(p_0)$ which passes through $x$ (equivalently a normal direction $y$ with $\langle y, x \rangle = p_0$ that defines a cap), exists that contains none of the samples. Due to the rotational invariance of the uniform distribution over the sphere, from which samples $x'$ haven been drawn, we can fix this cap and get

$$\mathbb{P}(p \leq p_0) \leq \mathbb{P}(X^1 \cap C_{p_0} = \emptyset) \tag{29}$$

where $C_{p_0} = \left\{ g \in \mathcal{S}^{d-1} \; : \; \langle g, x \rangle \geq p_0 \right\}$.

**Lemma 15.** *We have*

- *for $p_0 \leq \sqrt{\frac{2}{d}}$, we have $\frac{1}{12} \leq \mathbb{P}(C_{p_0}) \leq \frac{1}{2}$.*

- *for $\sqrt{\frac{2}{d}} \leq p_0 \leq 1$:*

$$\frac{1}{6p_0\sqrt{d}}(1 - p_0^2)^{(d-1)/2} < \mathbb{P}(C_{p_0}) < \frac{1}{2p_0\sqrt{d}}(1 - p_0^2)^{(d-1)/2}. \tag{30}$$

Using the above bound, and the independence of samples, we have $\mathbb{P}(p \leq p_0) = (1 - \mathbb{P}(C_{p_0}))^{N_1}$ which implies

$$\log \mathbb{P}(p \leq p_0) = N_1 \log\left(1 - \mathbb{P}(C_{p_0})\right) \leq -N_1 \mathbb{P}(C_{p_0}) < -N_1 \frac{1}{6p_0\sqrt{d}}(1 - p_0^2)^{(d-1)/2}$$

and gives the following bound when $d \geq 4$,

$$\mathbb{P}(p \leq p_0) < \exp\left(-N_1 \frac{1}{6p_0\sqrt{d}}(1 - p_0^2)^{(d-1)/2}\right). \tag{31}$$

Finally, we want the above probability to be small for $p_0 \leftarrow \frac{r_0}{\sqrt{1+r_0^2}}$; i.e. the following to be small

$$\sum_{i=1}^{k} N_1 \exp\left(-N_1 \frac{1}{6r_0\sqrt{d}}(1 + r_0^2)^{1-d/2}\right) \tag{32}$$

which can be insured when $\min_{i=1,\dots,k} N_i \geq 3$ by

$$\frac{1}{6r_0\sqrt{d}}(1 + r_0^2)^{1-d/2} > \max_{i=1,\dots,k} \frac{\log N_i}{N_i} = \frac{\log N_{\min}}{N_{\min}}. \tag{33}$$

Considering $r_0 = \tan\theta$ we have $\cos\theta = \frac{1}{\sqrt{1+r_0^2}}$ and $\sin\theta = \frac{r_0}{\sqrt{1+r_0^2}}$. Then, the above inequality can be expressed as

$$\frac{(\cos\theta)^{d-1}}{6\sqrt{d}\sin\theta} > \frac{\log N_{\min}}{N_{\min}}. \tag{34}$$

Simple manipulations yield the result of Theorem 5.

## C.3 Proof of Theorem 6: true positive rate when samples are drawn uniformly at random

In Theorem 4, given a value of $\epsilon_\rho$ determined by (deterministicly provided) samples, we established an upper bound for $\beta$ that can be used in $\mathrm{CSC}_1(\beta, x)$ to yield at least a fraction $\rho$ of samples as true positives. Under the random-sample model, $\epsilon_\rho$ is a random variable and $\arccos(\epsilon_\rho)$ follows a Beta distribution. Therefore, our upper bound for $\beta$ will be a random variable. In the following, we use this distributional information to provide an upper bound for $\beta$ which yields the desired $\rho$-rate *with high probability*.

Under the random-sample model, where samples $x'$ are drawn uniformly at random from the unit sphere in the given subspace, the quantity $\arccos(|\langle x', x\rangle|)$ has a uniform distribution $\mathrm{unif}([0, \frac{\pi}{2}])$; i.e.,

$$\arccos(|\langle x', x\rangle|) \sim \mathrm{unif}([0, \frac{\pi}{2}]).$$

We sort the values $\arccos(|\langle x', x\rangle|)$ for all $x' \in X$ and consider the $m$-th smallest value (as $\arccos$ is a decreasing on this domain). We denote this value by $\eta_\rho$ and note that it is the $m$-th order statistic for the uniform distribution $\mathrm{unif}([0, \frac{\pi}{2}])$. The $m$-th order statistic for the standard uniform distribution with $N-1$ samples obeys a Beta distribution, i.e.,

$$\frac{2}{\pi}\eta_\rho \sim \mathrm{Beta}(m, N - m),$$

with mean $\frac{m}{N}$ and variance $\frac{m(N-m)}{N^2(N+1)}$.

Recall that we defined $\epsilon_\rho = \cos(\eta_\rho)$. We use this distributional information to find a range for the value of interest (an upper bound $\beta_{U,\rho}^x$ on the $\beta$ which will be used for CSC to yield a fraction $\rho$ of true positives) with high probability.

**Lemma 16** (CDF for Beta distribution; e.g., see Section 2.2 of [13])**.** *For positive integer values $N$ and $m < N$, and for a Beta distributed random variable $Y \sim \mathrm{Beta}(m, N - m)$, with mean $\frac{m}{N}$ and variance $\frac{m(N-m)}{N^2(N+1)}$, we have:*

$$\mathbb{P}\left(Y < y\right) = I(y; m, N - m) := 1 - \sum_{i=1}^{m} \binom{N-1}{i-1} y^{i-1}(1 - y)^{N-i} \tag{35}$$

*where $I(y; m, N - m)$ is the* incomplete Beta function.

Using Lemma 16, we have

$$\frac{2}{\pi}\eta_\rho < \frac{m}{N} + \Delta \quad \text{with probability at least} \quad I(\frac{m}{N} + \Delta; m, N - m)$$

which implies $\epsilon_\rho = \cos(\eta_\rho) > \cos(\frac{\pi}{2}(\frac{m}{N} + \Delta))$ with high probability. Therefore, with high probability,

$$\beta_{u,\rho}^x = \frac{\sin^2\theta}{\cos\theta - \cos(\frac{\pi}{2}(\frac{m}{N} + \Delta))}$$

is the desired upper bound.

# D   True Positives for CSC vs SSC

In this section, we discuss a key feature of CSC, particularly as it relates to some prior work in subspace clustering. See [16] and references therein for a review of different approaches in subspace clustering. For concreteness and clarity of exposition, we contrast CSC with sparse subspace clustering (SSC) of [3] which is representative of the current state of the art. The first analysis of SSC appeared in [4] and was later improved in [14], and we use the latter in our discussions.

## D.1   True positive rate for SSC

In the following, we aim to illustrate an important distinction in the results of SSC and CSC by showing how the number of true positives of SSC is bounded by a quantity $|J(x)|$ which can be much smaller than the number of samples drawn from $S(x)$. More specifically, Lemma 17 tells us that the coefficients derived by the SSC optimization problem in (36) cannot be nonzero for samples that do not correspond to the extreme rays of $\mathcal{W}_N(x)$. In contrast, CSC faces no such limitation and for suitable values of $\beta$ one can have all true positives.

Notice that for a finitely generated cone, such as $\mathcal{W}_N(x) = \{(X - x\mathbf{1}_N^T)\lambda : \lambda \in \mathbb{R}_+^N\}$ defined in (4), the extreme rays are among the generators. Denote the set of extreme rays of $\mathcal{W}_N(x)$ by $\text{ext ray}\, \mathcal{W}_N(x)$. Then,

$$\text{ext ray}\, \mathcal{W}_N(x) = \text{cone}(Y) \quad \text{for some } Y \subseteq \{(X - x\mathbf{1}_N^T)e_i : i = 1, \ldots, N\}$$

where $e_i$ is the $i$-th standard basis vector in $\mathbb{R}^N$.

**Lemma 17.** *Given $X$, which has $N$ samples as its columns, consider an arbitrary column $x$ and the corresponding convex cone $\mathcal{W}_N(x) = \{(X - x\mathbf{1}_N^T)\lambda : \lambda \in \mathbb{R}_+^N\}$ defined in (4). Also, without loss of generality, assume that the columns of $X$ represent a symmetric set of points with respect to the origin. Consider any optimal solution $z^\star$ to*

$$\min_z \|z\|_1 \text{ subject to } Xz = x, \ z_x = 0, \ z_{-x} = 0. \tag{36}$$

*Then,*

$$z_i^\star \neq 0 \implies \text{sign}(z_i)x_i - x \in \text{ext ray}\, \mathcal{W}_N(x).$$

*In other words, when $X$ is symmetric, consider the subset of samples that are active in defining $\mathcal{W}_N(x)$ as*

$$J(x) := \{j : x_j \in (x + \text{ext ray}\, \mathcal{W}_N(x)) \cap \mathcal{S}^{n-1}, \ x_j \neq x\}. \tag{37}$$

*Then, for any optimal solution $z^\star$ to the above, we have $z_{J(x)^c}^\star = \mathbf{0}$.*

Notice that the above illustrates a very important point. While SSC, when successful, declares some of the members of $J(x)$ as being in the same subspace as $x$, CSC can potentially declare all the columns of $X$ in $S(x)$ as such. Therefore, *many more true positives are possible with CSC than with SSC*. In cases with independent subspaces, the true positive rate can go up as much as $\rho = 1$. Implications of this observation can be seen in the post-processing step for the recovered affinities, as well as in the proofs presented in [14].

*Proof of Lemma 17.* Suppose $x = x_1$, $x_2 \neq x_1$, and $z_2^\star \neq 0$. Since the $\ell_1$ penalty is invariant under sign flips and we are given a symmetric set of points, without loss of generality, suppose $z_2^\star > 0$ (otherwise we can flip the sign and still have an optimal solution, as $X$ is symmetric).

Contrapositively, assume $x_2 - x_1 \notin \text{ext ray } \mathcal{W}_N(x_1)$, which implies

$$\exists \lambda \geq 0 \;;\;\; x_2 - x_1 = \sum_{i=3}^{N} \lambda_i(x_i - x_1).$$

Setting $\lambda_1 = \lambda_2 = 0$, the above implies

$$x_2 = (1 - \mathbf{1}^T\lambda)x_1 + \sum_{i=3}^{N} \lambda_i x_i.$$

If $1 - \mathbf{1}^T\lambda \geq 0$ then $x_2$ is in the convex hull of $x_1, x_3, \ldots, x_N$, which contradicts the fact that all of the samples are on the unit sphere. Therefore, $\mathbf{1}^T\lambda > 1$.

Substituting for $x_2$ from the above equation in $x_1 = Xz^\star$ gives

$$x_1 = z_2^\star x_2 + \sum_{i=3}^{N} z_i^\star x_i = z_2^\star \left( (1 - \mathbf{1}^T\lambda)x_1 + \sum_{i=3}^{N} \lambda_i x_i \right) + \sum_{i=3}^{N} z_i^\star x_i$$

which gives

$$x_1 = \sum_{i=3}^{N} \frac{z_i^\star + z_2^\star \lambda_i}{1 + z_2^\star(\mathbf{1}^T\lambda - 1)} x_i$$

as another representation of $x_1$ in terms of the other columns of $X$; in addition to $x_1 = \sum_{i=2}^{N} z_i^\star x_i$. Recall $z_2^\star > 0$ and $\mathbf{1}^T\lambda > 1$. By optimality of $z^\star$ (as the vector of coefficients in representing $x_1$) we have

$$\|z^\star\|_1 \leq \sum_{i=3}^{N} \left| \frac{z_i^\star + z_2^\star \lambda_i}{1 + z_2^\star(\mathbf{1}^T\lambda - 1)} \right|$$

which gives

$$(1 + z_2^\star(\mathbf{1}^T\lambda - 1))\|z^\star\|_1 \leq \sum_{i=3}^{N} |z_i^\star + z_2^\star \lambda_i| \leq \sum_{i=3}^{N} |z_i^\star| + z_2^\star \sum_{i=3}^{N} \lambda_i = \|z^\star\|_1 + z_2^\star(\mathbf{1}^T\lambda - 1)$$

which gives $\|z^\star\|_1 \leq 1$ implying that $x_1$ lies in the convex hull of $\{\pm x_2, \ldots, \pm x_N\}$; which contradicts with the fact that all of these points are on the unit sphere. This establishes the claim that $z_2^\star \neq 0$ implies $x_2 - x_1 \in \text{ext ray } \mathcal{W}_N(x_1)$. $\qquad\square$

### D.2 Comparisons on true positives for CSC, SSC, and TSC

As illustrated by Lemma 17, the number of true positives for SSC is limited by the number of extreme rays of the corresponding tangent cones. While SSC has this fundamental limitation, the proposed conic geometry provides us with a way to address this issue with provable guarantees. Other methods with lots of true positives are fundamentally different as they use a different guarantee strategy or offer no guarantees at all. In contrast, the number of true positives for CSC is tunable by $\beta$ (via the characterizations we provide). While this number is nonetheless restricted to an admissible range, the range's upper limit could grow as large as the number of samples in each subspace; an easy example to observe this would be the case of independent subspaces (when the sum of dimensions is equal to the dimension of the Minkowski sum).

Furthermore, the distribution of true positives is favorable for CSC. More specifically, as we argue in the Section D.3 and illustrate via some numerical experiments, the true positives are distributed in a way that only $O(\log N_t)$ true positives per sample are enough to ensure graph connectivity in the random-sample model. This small value, provides us with a freedom as described in the following. In complicated situations where

- the inner radius of the tangent cone is small (for example, when *samples* are concentrated rather than uniformly spread over the subspace),
- or $\beta_L$ is large (i.e., when *subspaces* have a complicated configuration),

having a nonempty range for $\beta$ requires a large $\beta_{U,\rho}$ which corresponds to admissibility of only small $\rho$'s. In such case, the knowledge that true positives are favorably spread (as it is the case for CSC) enables us to run CSC expecting a very small number of true positives yet being sure about the final clustering results (because of true positives being well spread). On the other hand, with SSC, it is an open question to understand the spread of true positives, which depends on the complicated facial geometry of the data hull.

Subspace clustering via thresholding (TSC) [7] considers the $q$ closest points to a sample $x$ (and $-x$) to set their affinity with $x$ equal to 1. They propose using $\log d_t \ll q \ll d_t$. Unlike TSC, which only considers the distance between samples (a pure nearest-neighbors strategy), CSC relies on both the distances and the elongation of the cone in the corresponding plane defined by each pair of samples. Furthermore, the true positives pattern is different between CSC and TSC: because TSC uses $q \ll d_t$, it may face connectivity issues. Finally, the choice of $q$ is fixed for all samples and cannot adapt to the dimension of the corresponding subspace, the configuration of the subspaces, or the density of samples from each subspace. Hence, such nearest-neighbor-based method can be easily misled by imbalance in the subspace dimensions and their sample densities.

### D.3 True positives pattern for CSC

We restrict our attention to a single subspace of dimension $d$ and consider a random-sample model where $N$ samples are drawn uniformly at random. Fix a reference sample $x$. We will use the notations and definitions from Appendix C. Assume, with high probability: given the samples, the inner radius of the intersection of shifted tangent cone with $\{x\}^\perp$ is $r$ (corresponding to a value $\theta$ in Theorem 4). To compute $\beta_{U,\rho}^x$, use the $m$-th largest absolute inner product; $m \coloneqq \lceil \rho(N-1) \rceil$, and the above inradius. Then any sample $x'$ with $|\langle x, x' \rangle| > \epsilon_\rho$ will certainly be among the true positives for $\mathrm{CSC}_1(\beta, x)$ for $\beta \leq \beta_{U,\rho}^x$. Hence, the following matrix

$$A_{\mathrm{sub}} \overset{\mathrm{dist}}{\sim} \text{keep top } m \text{ values in each row of } |X^T X|, \quad X \sim \mathtt{normc}(\mathcal{N}(0, I_d \otimes I_N))$$

will have a support *inside* the support of the affinity matrix constructed by $\mathrm{CSC}_1(\beta)$ where $\beta \leq \beta_{U,\rho}$ and $\beta_{U,\rho}$ is defined via the conditioned value of inradius and directly using the $m$-th order statistic. In the above, we use MATLAB's notation $\mathtt{normc}$ to denote a matrix derived by normalizing each column of the input matrix. If we use the ingredients from proof of Theorem 6, we can bound the order statistic with high probability, hence using

$$A_{\mathrm{sub}} \overset{\mathrm{dist}}{\sim} \left(|X^T X| \geq \epsilon\right), \quad X \sim \mathtt{normc}(\mathcal{N}(0, I_n \otimes I_N))$$

as the matrix whose support is included in the support of the affinity matrix constructed by CSC, with high probability. In this setting, the only question that remains is to characterize when (for which $n$, $N$ and $\rho$) is the graph corresponding to $A$ *disconnected*?

The connectivity of the graph corresponding to $A_{\mathrm{sub}}$ can be studied empirically thanks to the above distributional characterization. We considered all the combinations of subspace dimension ($d$) and number of samples per dimension ($\kappa = N/d$) from $d \in \{20, 40, 60, \ldots, 200\}$ and $\kappa \in \{1.2, 1.6, 2, \ldots, 8\}$, and generated 100 different matrices $X \in \mathbb{R}^{d \times \kappa d}$, with i.i.d. normal entries normalized to have unit $\ell_2$ norm columns, for each pair of values $(d, \kappa)$. Here $N = \kappa d$. Then, for each $X$, using a bisection approach with 15 steps, we found the smallest $\rho$ for which the graph corresponding to $A_{\mathrm{sub}}$ is connected. We denote the average value of such minimal $\rho$, over the 100 trials, by $\rho_{\mathrm{con}}(d, \kappa)$. We observe that $\rho_{\mathrm{con}}$ is tightly approximated by $1.1 \frac{\log(\kappa d)}{\kappa d}$, and Figure 9 provides the multiplicative residual for different values of $\kappa$ and $d$. Recall that $p = \frac{\log N}{N}$ is a sharp threshold for connectivity of an Erdos-Renyi graph $\mathcal{G}(N, p)$. While the adjacency matrix for our random graph does not have i.i.d. Bernoulli($p$) entries as an Erdos-Renyi graph does, its connectivity threshold shows a strong resemblance with the threshold for Erdos-Renyi graphs.

The above observation allows us to be confident in perfect clustering even if $\theta_t$ (dictated by the number of samples) and the formula for computing the upper bound (which has to be larger than $\beta_L$) only allow us to work with small values of $\rho$: even if the formulas only allow for logarithmically small values of $\rho$, we can still use CSC and expect perfect clustering.

Figure 9: Scaling of $\rho_{\mathrm{con}}(d, \kappa) \simeq 1.1 \frac{\log(\kappa d)}{\kappa d}$ which makes the thresholded graph connected. Shade of each pixel corresponds to $\frac{\rho_{\mathrm{con}}(d,\kappa)}{\log(\kappa d)/\kappa d}$.

# E Details on Equivalent Formulations in Section 7

Taking the dual of linear feasibility program (CR), the *cone membership (CM)* can be tested by solving

$$\min_{y \in \mathbb{R}^n} \ \langle y, \beta x' - x \rangle \quad \text{subject to} \quad y^T(X - x\mathbf{1}_N^T) \geq 0 \tag{CM}$$

where $[\![(\mathrm{CM}) : 0, \mathrm{unbounded}]\!]$. This program checks whether there exists a certificate $y \in \mathcal{W}_N^\star(x)$ (in the dual cone) that rejects the membership of $\beta x' - x$ in $\mathcal{W}_N(x)$. Notice that neither (CR) nor (CM) are robust and their output changes from zero with the smallest departure of $\beta x' - x$ from the cone. On the other hand, observe that we are reading off very little information from these optimization programs (feasibility or boundedness) and this provides us with an opportunity to tweak these optimization problems without changing the categorization offered by $[\![(\mathrm{CR}) : 0, \mathrm{infeasible}]\!]$ and $[\![(\mathrm{CM}) : 0, \mathrm{unbounded}]\!]$, while coming up with more robust programs. Next, we discuss a proposal to turn this unbounded feasibility problem into a *bounded feasible* linear program with outputs 0 and 1.

**Reformulations for faster and more robust optimization.**   Observe that restricting $y$ to any set with origin in its relative interior yields a program $P$ that is in CR-class, with $[\![P : 0, \mathrm{negative}]\!]$. For example, consider an augmentation to (CM) as

$$\min_{y \in \mathbb{R}^n} \ \langle y, \beta x' - x \rangle \quad \text{subject to} \quad y^T(X - x\mathbf{1}_N^T) \geq 0 \,, \ \langle y, \beta x' - x \rangle \geq -\epsilon \tag{TCM}$$

for some $\epsilon > 0$, which we refer to as the *truncated cone membership (TCM)* program. Clearly, the program is now feasible and bounded and we have $[\![(\mathrm{TCM}) : 0, -\epsilon]\!]$. Furthermore, this program can be solved approximately, up to a precision $\epsilon' \in (0, \epsilon)$, and provide the same desired set of results, i.e., $[\![\epsilon'\text{-inexact (TCM)} : \mathrm{nonnegative}, \mathrm{negative}]\!]$: the $\epsilon'$-inexact solution is nonnegative if and only if $\beta x' - x \in \mathcal{W}_N(x)$. Since $\epsilon$ can be chosen arbitrarily, this formulation allows us to choose the desired bit-length of the solution and optimize the overall computational cost of optimization.

If we dualize (TCM) and divide the objective by $-\epsilon$ we get the *robust cone membership (RCM)* program,

$$\min_{\gamma \geq 0, \lambda \geq 0} \ \gamma \quad \text{subject to} \quad (1 - \gamma)(\beta x' - x) = (X - x\mathbf{1}_N^T)\lambda \tag{RCM}$$

where $\gamma$ is a scalar and $\lambda \in \mathbb{R}_+^N$. This problem can similarly be written by tweaking the conic representation problem (CR) and mapping 0 and $-\infty$ to 0 and 1; without double dualization. However, the duality relationship with (TCM) is helpful in understanding the dual space and devising efficient optimization algorithms. Notice that $(\gamma, \lambda) = (1, 0_N)$ is always feasible, and the optimal solution is in $[0, 1]$. Moreover, $[\![(\mathrm{RCM}) : 0, 1]\!]$, which makes it a desirable candidate as a proxy for $x' \notin S(x)$. We use this program in our experiments reported in Section 3.1.

Moreau's decomposition theorem [9] can be stated as: For a given convex cone $\mathcal{W}$, we have $y \in \mathcal{W}$ if and only if $\Pi(y; \mathcal{W}^\circ) = \mathbf{0}$, where $\mathcal{W}^\circ \coloneqq \{z : \langle z, w \rangle \leq 0, \, \forall\, w \in \mathcal{W}\}$ is the polar cone. Using this, we can solve the following optimization program, tagged as the *dual cone projection* (DCP) program,

$$\min_{y \in \mathbb{R}^n} \quad \langle y, \beta x' - x \rangle + \frac{1}{2}\|y\|_2^2 \quad \text{subject to} \quad y^T(X - x\mathbf{1}_N^T) \geq 0 \qquad \text{(DCP)}$$

and $[\![(\text{DCP}) : 0, [-\frac{1}{2}\|\beta x' - x\|_2^2, 0)]\!]$. Observe that, when the problem is feasible, it is always bounded, which is a very useful property in using optimization algorithms. Adding a constant term $\frac{1}{2}\|\beta x' - x\|_2^2$ to (DCP) illustrates its equivalence to a projection onto the dual cone, and gives

$$\frac{1}{2}\text{dist}^2(-\beta x' + x, \mathcal{W}_N^\star(x)).$$

This illustrates that (DCP) provides a reasonable proxy for the *degree of membership* and can be used to define a continuous distance function (as opposed to 0/1 based on membership) that is also more robust to noisy samples.

Aside from the computational and robustness properties of these reformulations, one can study their relationship with geometry of the set of data points. Below, we mention three remarks.

*Different bases of the cone.* An inhomogenous version of (CM) can be derived by replacing $y \in \mathcal{W}_N^\star(x)$ with one that insures $y$ is in a *base* of $\mathcal{W}_N(x)$. Since $\mathcal{W}_N(x)$ is pointed and full-dimensional, and contains $-x$, the dual cone is also pointed and is included in $\{y : \langle y, x \rangle \leq 0\}$. Therefore, the intersection of the cone with $\langle y, x \rangle = 1$ is a base for $\mathcal{W}_N^\star(x)$. Interestingly, this set is equal to $-\partial\|x\|_X$; when $X$ contains symmetric data points (with origin in the interior of its convex hull) and its symmetric gauge function gives a norm, denoted by $\|\cdot\|_X$. Moreover, the optimal solution to this program is either zero (when in the cone) or the negative of the directional derivative;

$$-\max_{g \in \partial\|x\|_X} \quad \langle g, \beta x' - x \rangle. \qquad (38)$$

*Data norm.* The constrained program (CM) can be turned into an unconstrained form, by regularizing for the constraints, as

$$\min_{y \in \mathbb{R}^n} \quad \langle y, \beta x' - x \rangle + \theta\left(\|X^T y\|_\infty + \langle y, x \rangle\right) \qquad (39)$$

which through convex conjugacy is equivalent to checking whether $\frac{\beta}{\theta}x' + (1 - \frac{1}{\theta}x)$ is in $\text{conv}([-X, X])$ or not. The latter is in turn equivalent to checking whether $\beta x' - x$ is a descent *direction* at $x$ with respect to $\|\cdot\|_X$.

*A notion of self-expressiveness.* We can stack all the optimization problems in (RCM) (also used in $\text{CSC}_1(\beta, x, x')$) for each $x$ and scalarize the resulting multi-objective optimization problem and solve

$$\hat{\gamma}(x) = \min_{\gamma \geq 0, \Lambda \geq 0} \quad \|\gamma\|_2^2 \quad \text{subject to} \quad \beta X \text{diag}(1 - \gamma) - x\gamma^T = (X - x\mathbf{1}_N^T)\Lambda$$

where $\gamma \in \mathbb{R}^N$ and $\Lambda \in \mathbb{R}^{n \times N}$. Notice that the problem is completely separable in the entries of $\gamma$ and the columns of $\Lambda$, hence equivalent to (RCM). Moreover, one can use any other separable function instead of the $\|\gamma\|_2^2$ in the above. While, in general, any data point can be written as a combination of other data points in the space they span, it is interesting to understand the properties of the specific expression given in the constraints of the above optimization problem. In fact, self-expressiveness property is a rather generic and vague description. For example, CSC can be written as a self-expressive representation with coefficients from an extended simplex. $k$-means (with a subset of samples as centers) can also be written as a self-expressive representation with a 0/1 coefficient matrix with $k$ nonzero rows, and with only one 1 in each column.

### E.1 Computational complexity and practical considerations

The proposed tests for CSC consist of solving $O(N^2)$ optimization problems, for each pair of $x, x'$, each of which in $n$ variables and with $N$ constraints. Different methods can be adopted to solve these linear feasibility programs or the variations. However, there are multiple ways we might be able to get the same final output affinity matrix with much less computation.

*Inexact solutions are enough.* since the goal in (CM) or (DCP) is distinguishing between bounded (with zero optimal value) and unboundedness of the problem, one only needs to solve the problem until reaching any negative value for the objective function. We have discussed an idea along this line when we introduced (ITCM).

*Partial examination is enough.* suppose we are guaranteed to have no false positives under certain subspace configurations and sampling scenarios. This means that in the union of cliques graph, with nodes being the samples, we only need to find a connected subgraph, which can then be completed to the union of cliques graph. Therefore, in such cases, we only need to solve $O(N)$ optimization problems instead of $O(N^2)$ ones. More concretely, as discussed in Section D, CSC provides a rather well-spread approximation of the original graph (which is a union of $k$ cliques corresponding to samples from each subspace). Therefore, solving a subset of $O(N^2)$ problems would still allow for perfect grouping by the spectral clustering (the post-processing) step.

It is worth mentioning that SSC requires solving $N$ $\ell_1$-minimization problems with $N$ variables and $n$ constraints. However, when the number of samples grow compared to the dimension of each subspace, the output affinity matrix by SSC becomes very sparse (due to the limitation in its design, characterized in Section D). Therefore, in essence, SSC becomes *inapplicable* when we require a large number of samples, e.g., when the subspaces are highly aligned and a higher number of samples is expected to make the subspaces recoverable.

# F  Background from Inversive Geometry

*Inversion* [1] is a geometric transformation which preserves angles (but reverses the direction, hence is anti-conformal). The inverse of a line or a circle in the plane is a line or a circle in the plane. We will use inversion to understand a mapping from $\{x' :\ x' \in S(x) \cap X\}$ to $(x + \mathcal{W}_N^1(x)) \cap \{x\}^\perp$.

Consider an inversion map centered at $x$ and with radius $\sqrt{2}$; denoted by

$$I(\cdot) = I_{x,\sqrt{2}}(\cdot) \tag{40}$$

where

$$I_{x,r}(y) := x + \frac{r^2}{\|y - x\|_2^2}(y - x) = \frac{r^2}{\|y - x\|_2^2}y + \left(1 - \frac{r^2}{\|y - x\|_2^2}\right)x. \tag{41}$$

Moreover, for a given point $x \in \mathcal{S}^{d-1}$, consider two other operators as in the following:

- *Radial Projection* defined as $\mathring{\Pi}_x : \mathcal{S}^{d-1}\backslash\{x\} \to \{x\}^\perp$ which maps a point $g$ on the unit sphere to the intersection of the line $\{\alpha x + (1 - \alpha)g :\ \alpha \in \mathbb{R}\}$ and the hyperplane $\{x\}^\perp$. More concretely,

$$\mathring{\Pi}_x(g) = \frac{g - \langle g, x \rangle x}{1 - \langle g, x \rangle}. \tag{42}$$

- *Radial Lift* defined as $\mathring{\amalg}_x : \{x\}^\perp \to \mathcal{S}^{d-1}\backslash\{x\}$ which maps a point $h$ on the hyperplane $\{x\}^\perp$ to the intersection of the line $\{\alpha x + (1 - \alpha)h :\ \alpha \in \mathbb{R}\}$ and the unit sphere other than $x$ itself. More concretely,

$$\mathring{\amalg}_x(h) = \frac{2}{\|h\|_2^2 + 1}h + \frac{\|h\|_2^2 - 1}{\|h\|_2^2 + 1}x. \tag{43}$$

It is easy to verify that $\mathring{\Pi}_x$ and $\mathring{\amalg}_x$ are inverse maps, and $I_{x,r}$ is its own inverse; i.e.

$$\mathring{\Pi}_x\mathring{\amalg}_x(g) = g \quad \text{for all } g \in \mathcal{S}^{d-1}\backslash\{x\}, \tag{44}$$

$$\mathring{\Pi}_x\mathring{\amalg}_x(h) = h \quad \text{for all } h \in \{x\}^\perp, \tag{45}$$

$$I_{x,r}I_{x,r}(y) = y \quad \text{for all } y \in \mathbb{R}^d\backslash\{x\}. \tag{46}$$

In the following, we review some other useful properties of these operators.

**Some preliminary properties.** As mentioned before, the inverse of a line or a circle is a line or a circle. Moreover, as clear from the definition, a point and its inverse lie on the same line passing through the center of inversion. We understand the inversion mapping as a mapping defined over all closed subsets of $\mathbb{R}^d \setminus \{x\}$ through

$$A = \bigcup_{i \in \mathcal{I}} (A \cap \ell_i) \; : \; I_{x,r}(A) = \bigcup_{i \in \mathcal{I}} I_{x,r}(A \cap \ell_i) \tag{47}$$

where $\{\ell_i \; : \; i \in \mathcal{I}\}$ is a collection of lines (we could alternatively use circles) covering the whole $\mathbb{R}^d$.

**Lemma 18.** *Consider a hyperplane $H \subset \mathbb{R}^d$ which passes through $x$. Then,*

$$I_{x,\sqrt{2}}(H) = H. \tag{48}$$

*Proof.* $H$ can be understood as a collection of lines that are all passing through the origin. The inverse of each such line is itself, yielding the claim. $\square$

Later, when we consider convex sets, we understand them as the intersection of halfspaces. To be able to use the neat properties of inversion, we need to understand the inverse of hyperplanes that define these halfspaces.

**Corollary 19.** *Consider a hyperplane $H \subset \mathbb{R}^d$ which passes through $x$. Then,*

$$I_{x,\sqrt{2}}(H \cap \mathcal{S}^{d-1}) = H \cap \{x\}^{\perp}. \tag{49}$$

*Proof.* It is easy to see that

$$I_{x,\sqrt{2}}(H \cap \mathcal{S}^{d-1}) = I_{x,\sqrt{2}}(H) \cap I_{x,\sqrt{2}}(\mathcal{S}^{d-1}) = H \cap \{x\}^{\perp}.$$

$\square$

**Lemma 20.** *We have $I_{x,\sqrt{2}}(\{x\}^{\perp}) = \mathcal{S}^{d-1} \setminus \{x\}$.*

*Proof.* Observe that the inverse of $-x$ is the origin;

$$I_{x,\sqrt{2}}(-x) = \mathbf{0}.$$

Consider the unit sphere as the union of its 2-dimensional equators passing through $x$ and $-x$. In other words, consider all possible shortest paths on the sphere from $x$ to $-x$. Since these circles pass through the center of inversion, i.e. $x$, their inverses are straight lines not passing through $x$. However, they all pass through $-x$ and their inverse will contain the inverse of $-x$, namely the origin. Moreover, the line $[-x, x]$ is orthogonal to all these circles which implies that all of the inverses will be orthogonal to the inverse of $[-x, x]$, namely the line itself. All in all, the inverse of the unit sphere is a subset of $\{x\}^{\perp}$. Following similar arguments yields equality. $\square$

**Lemma 21.** *Consider two symmetric points $g, -g$ on the unit sphere, different from the center of inversion $x$. Then the origin lies on the line segment connecting $\mathring{\Pi}_x(g)$ and $\mathring{\Pi}_x(-g)$.*

*Proof.* There are multiple ways to establish this statement. First, since $g, -g$ are symmetric with respect to the origin, the sphere has a great circle passing through $-g, -x, g, x$. The inverse of this circle is a line in $\{x\}^{\perp}$ passing through the origin. Considering the half-circle from $-g$ to $-x$ to $g$, the inverse is a line from $\mathring{\Pi}_x(-g)$ to $\mathring{\Pi}_x(-x) = 0$ to $\mathring{\Pi}_x(g)$ which establishes the claim.

Alternatively, we can show that the following equality is feasible with some value of $\lambda \in [0, 1]$;

$$\lambda \mathring{\Pi}_x(g) + (1 - \lambda) \mathring{\Pi}_x(-g) = 0.$$

In fact, plugging in the expression for $\mathring{\Pi}_x(\cdot)$ from (42), after some algebraic manipulations, we get

$$(g - x\langle x, g \rangle)(1 - \langle x, g \rangle - 2\lambda) = 0$$

Since $\pm g \neq x$, the first term is nonzero; take the inner product with $g$. Therefore, we get $\lambda = \frac{1}{2}(1 - \langle x, g \rangle) \in [0, 1]$ which establishes the claim. $\square$

All in all, we get the following:

**Lemma 22.** *The restriction of the inversion operator, centered at $x$ and with radius $\sqrt{2}$, to the unit sphere is equivalent to the radial projection operator; i.e.,*

$$\mathring{\Pi}_x(g) \equiv I_{x,\sqrt{2}}(g) \quad \text{for all } g \in \mathcal{S}^{d-1} \backslash \{x\}. \tag{50}$$

*Furthermore, the restriction of the inversion operator, centered at $x$ and with radius $\sqrt{2}$, to $\{x\}^{\perp}$ is equivalent to the radial lift operator; i.e.,*

$$\mathring{\Pi}_x(h) \equiv I_{x,\sqrt{2}}(h) \quad \text{for all } h \in \{x\}^{\perp}. \tag{51}$$

**A convex hull in $\{x\}^{\perp}$.** Suppose $x \in S_t$, only for $t = 1, \ldots, j$. Observe that

$$(x + \mathcal{W}_N^{1,\ldots,j}(x)) \cap \{x\}^{\perp} = \mathring{\Pi}_x((x + \mathcal{W}_N^{1,\ldots,j}(x)) \cap \mathcal{S}^{d-1}) \tag{52}$$

$$= \operatorname{conv} \mathring{\Pi}_x \left( \{x' : x' \in S(x)\} \right). \tag{53}$$

Therefore, the inradius of $(x + \mathcal{W}_N^{1,\ldots,j}(x)) \cap \{x\}^{\perp}$ is equal to the minimum distance of the origin to all faces of $\operatorname{conv} \mathring{\Pi}_x \left( \{x' : x' \in S(x)\} \right)$.

The convex set $\operatorname{conv} \mathring{\Pi}_x \left( \{x' : x' \in S(x)\} \right) \subset \{x\}^{\perp}$ can be characterized as the intersection of all of its supporting halfspaces. Each of these halfspaces: 1) can be radially lifted to a halfspace passing through $x$, 2) contains the origin, as implied by Lemma 21, when $X$ is symmetric. We index these subspaces by a set $\mathcal{I}$, and for each $i \in \mathcal{I}$, we denote the halfspace defined by the hyperplane $H_i$ and containing the origin by $H_i^{\leq 1}$. Therefore, the inverse of this convex hull (denote by $C$) can be characterized as

$$I(C) = I(\bigcap_{i \in \mathcal{I}} H_i^{\leq 1} \cap \{x\}^{\perp}) = \bigcap_{i \in \mathcal{I}} I(H_i^{\leq 1}) \cap \mathcal{S}^{d-1} = \bigcap_{i \in \mathcal{I}} H_i^{\leq 1} \cap \mathcal{S}^{d-1}$$

which is the unit sphere minus a union of caps passing through $x$; see Figure 10. Moreover, observe that $\bigcap_{i \in \mathcal{I}} H_i^{\leq 1} = x + \mathcal{W}_N^{1,\ldots,j}(x)$.

Figure 10: The radial lift of a convex set in $\{x\}^{\perp}$ (shown with thick black lines) is the unit sphere minus a number of caps defined by the supporting hyperplanes of the convex set that also pass through $x$. The radially lifted set if shown in green.