[Reviews · NeurIPS 2017]

Reviewer 1



The paper has been already published and presented at FOCM2017.

Reviewer 2



Summary: The authors propose a new approach for clustering high dimensional data points according to subspace membership that they call “conic subspace clustering” (CSC). The new approach is based on an intuitive geometric relationship between the subspace containing a given data point and its tangent cone relative to all other data points. The authors leverage this relationship to devise a membership test that can be carried out via a sequence of linear programs. Rigorous analysis of the method is established to guarantee that the test succeeds under distributional assumption, as well as motivate parameter choices used in implementation. Some small-scale numerical results show that the method works better than the well-known sparse subspace clustering (SSC) algorithm based on sparse approximation. Strengths: - The contributions of the paper to the literature are significant. - The method is intuitive, and qualitatively seems to fix issues that have been hindering SSC-type methods that rely on accurate construction of an affinity matrix, followed by spectral clustering. - The authors introduce several new geometrical concepts and tools for the subspace clustering problem that do not seem to have been utilized before in the literature, and could potentially be of interest in other problems as well. Weaknesses: - The main paper is dense. This is despite the commendable efforts by the authors to make their contributions as readable as possible. I believe it is due to NIPS page limit restrictions; the same set of ideas presented at their natural length would make for a more easily digestible paper. - The authors do not quite discuss computational aspects in detail (other than a short discussion in the appendix), but it is unclear whether their proposed methods can be made practically useful for high dimensions. As stated, their algorithm requires solving several LPs in high dimensions, each involving a parameter that is not easily calculable. This is reflected in the authors’ experiments which are all performed on very small scale datasets. - The authors mainly seem to focus on SSC, and do not contrast their method with several other subsequent methods (thresholded subspace clustering (TSC), greedy subspace clustering by Park, etc) which are all computationally efficient as well as come with similar guarantees.

Reviewer 3



I thank the authors for their feedback. I agree that the paper contains interesting new ideas. But for a well-established problem such as subspace clustering, I find it important to put the new contributions clearly into context. Overall, I have increased my score from 4 to 5. Original review: The paper studies a new approach to subspace clustering. The main idea is a new criterion for determining the "affinity", i.e., a measure for how likely it is that two points belong to the same subspace (the resulting affinity matrix is then usually clustered with spectral methods). The proposed criterion is based on whether certain test points (depending on the two points for which we compute the affinity) can be expressed as the conic combination of the datapoints. While I find the proposed approach interesting, the comparison to prior work is somewhat incomplete. Sparse subspace clustering (where the datapoints are represented as a sparse combination instead of a conic combination) is a well-established method in this field. The paper refers to sparse subspace clustering many times, but the main comparison in terms of theoretical results is deferred to the appendix. The focus of the comparison is also mainly on the true positive rate, not the final clustering objective. Do the authors expect that their proposed method also improves in the final clustering objective? On the empirical side, the authors compare to sparse subspace clustering, but only on a small synthetic example. It would be interesting to see how the methods compare on somewhat larger synthetic test cases and on real data. Overall, I find the proposed ideas intriguing and would like to see how they compare to existing methods in subspace clustering. Unfortunately, I find the current paper not thorough enough in that respect. If the current version of the paper is rejected, I encourage the authors to submit a revised version with a more detailed comparison to a venue at the same level as NIPS.

Reviewer 4



Summary: The paper describes a new geometric insight and a new algorithm for noiseless subspace clustering based on a geometric insight about the tangent cone. The key idea (Proposition 2) is that if x and x’ are from the same subspace, then -x -\sign(< x,x’ >) \beta x’ should within a union of non-convex cones that comes directly from x and the underlying union of subspaces for all positive \beta, while if x and x’ are not from the same subspace, then when \beta is sufficiently large then the vector will not be on this union of non-convex cones. The authors then propose an algorithm that tests every ordered pair of observed data points with a similar condition for a crude approximation of the union of non-convex cones’’ of interest ---- on the conic hull (therefore convex) of the observed data points. The algorithm is parameterized by a tuning parameter \beta. The authors then provide geometric conditions under which: (1). if \beta is sufficiently large then there are no false positives. (2) if \beta is sufficiently small, then the true positive rate is at least a fixed fraction. The geometric conditions are different for (1) and (2). For (1) the condition is reasonable, but for (2) it is quite restrictive, as I will explain later. In addition, the authors analyzed the method under semi-random model (uniform observations on each subspace, or uniformly random subspaces themselves ) and described how to control the true positive rate. Evaluation: The paper is well-written. The technical results appear to be correct (didn’t have time to check the proofs in details). The idea of testing the “affinity” of pairs of data points using tangent cone is new as far as I know. The true positive rate control in Section 4 is new and useful. The result itself (in Section 4 and 5) seems quite preliminary are has a number of issues. In particular, it is not clear under what condition there exists a \beta that simultaneously satisfy conditions in Theorem 4 and Theorem 7. Or rather that the largest \rho one can hope to get when we require \beta > \beta_L, and how that number depends on parameters of the stochastic model. On the other hand, the proposed approach does not seem to solve a larger class of problems than known before through using SSC. Specifically, SSC allows the number of subspaces to be exponential in the ambient dimension, while the current approach seems to require the sum of dimensions of all subspaces to be smaller than the ambient dimension n. This is critical and not an artifact of the proof because in that case, the tangent cone would be R^n (or is it a half space?). (For example, consider three 1D disjoint subspaces in 2D). Part of the motivation of the work is to alleviate the graph connectivity’’ problem that SSC suffers. I believe the authors are not aware of the more recent discoveries on this front. This issue is effectively solved the noiseless subspace clustering problem with great generality [a]. Now we know that if SSC provides no false discoveries then there is a post processing approach that provides the correct clustering despite potentially disconnected graph for data points in the subspace. The observation was also extended to noisy SSC. The implication of this to the current paper (which also construct an affinity graph) is that as long as there are no false discoveries and at least d true discoveries (d is the subspace dimension) then the correct clustering can be obtained. Some discussions of the advantage of the proposed approach to the robust post-processing approach described in [a]. Lastly, since the proposed method is based on a seemingly very unstable geometric condition, I feel that it is much more important to analyze the robustness to stochastic noise as in [b,15] and adversarial perturbation as in [a,b]. Overall, I feel that this paper contains enough interesting new insights that should merit an accept to NIPS. Detailed comments/pointers and references: 1. Noiseless subspace clustering is, in great generality, a solved problem. - For l1 SSC, there is a postprocessing approach that provides perfect clustering under only identifiability condition as long as there is no false positives. - In other word, on noiseless subspace clustering problem, SSC does not suffer from the graph connectivity problem at all. - As a result, the authors should explicitly compare the geometric condition used in this paper to the condition for SSC to yield no-false discoveries [14]. 2. Even those geometric conditions in [14] can be removed if we allow algorithms exponential in subspace dimension d (and polynomial in everything else). In fact, L0 verson of SSC can already achieve that solve all subspace clustering problems that are identifiable [a]. 3. The noisy subspace clustering and missing data subspace clustering problems are where many of the remaining open problems are. So it will be great if the authors can continue to pursue the new tangent cone idea and figure out how they can solve the noisy/missing data problems in comparison to those approaches existing in the literature. [a] Wang, Yining, Yu-Xiang Wang, and Aarti Singh. "Graph connectivity in noisy sparse subspace clustering." Artificial Intelligence and Statistics. 2016. [b] Wang, Yu-Xiang, and Huan Xu. "Noisy Sparse Subspace Clustering." Proceedings of the 30th International Conference on Machine Learning (ICML-13). 2013.